# Human alignment of neural network representations

**Lukas Muttenthaler, Jonas Dippel, Lorenz Linhardt, Robert A. Vandermeulen**
Machine Learning Group, Technische Universität Berlin & BIFOLD
Berlin, Germany

**Simon Kornblith**
Google Research, Brain Team

## Abstract

Today's computer vision models achieve human or near-human level performance across a wide variety of vision tasks. However, their architectures, data, and learning algorithms differ in numerous ways from those that give rise to human vision. In this paper, we investigate the factors that affect the alignment between the representations learned by neural networks and human mental representations inferred from behavioral responses. We find that model scale and architecture have essentially no effect on the alignment with human behavioral responses, whereas the training dataset and objective function both have a much larger impact. These findings are consistent across three datasets of human similarity judgments collected using two different tasks. Linear transformations of neural network representations learned from behavioral responses from one dataset substantially improve alignment with human similarity judgments on the other two datasets. In addition, we find that some human concepts such as food and animals are well-represented by neural networks whereas others such as royal or sports-related objects are not. Overall, although models trained on larger, more diverse datasets achieve better alignment with humans than models trained on ImageNet alone, our results indicate that scaling alone is unlikely to be sufficient to train neural networks with conceptual representations that match those used by humans.

## 1 Introduction

Representation learning is a fundamental part of modern computer vision systems, but the paradigm has its roots in cognitive science. When Rumelhart et al. (1986) developed backpropagation, their goal was to find a method that could learn representations of concepts that are distributed across neurons, similarly to the human brain. The discovery that representations learned by backpropagation could replicate nontrivial aspects of human concept learning was a key factor in its rise to popularity in the late 1980s (Sutherland, 1986; Ng & Hinton, 2017). A string of empirical successes has since shifted the primary focus of representation learning research away from its similarities to human cognition and toward practical applications. This shift has been fruitful. By some metrics, the best computer vision models now outperform the best individual humans on benchmarks such as ImageNet (Shankar et al., 2020; Beyer et al., 2020; Vasudevan et al., 2022). As computer vision systems become increasingly widely used outside of research, we would like to know if they see the world in the same way that humans do. However, the extent to which the conceptual representations learned by these systems align with those used by humans remains unclear.

Do models that are better at classifying images naturally learn more human-like conceptual representations? Prior work has investigated this question indirectly, by measuring models' error consistency with humans (Geirhos et al., 2018; Rajalingham et al., 2018; Geirhos et al., 2021) and the ability of their representations to predict neural activity in primate brains (Yamins et al., 2014; Güçlü & van Gerven, 2015; Schrimpf et al., 2020), with mixed results. Networks trained on more data make somewhat more human-like errors (Geirhos et al., 2021), but do not necessarily obtain a better fit to brain data (Schrimpf et al., 2020). Here, we approach the question of alignment between human and machine representation spaces more directly. We focus primarily on human similarity judgments

collected from an odd-one-out task, where humans saw triplets of images and selected the image most different from the other two (Hebart et al., 2020). These similarity judgments allow us to infer that the two images that were not selected are closer to each other in an individual's concept space than either is to the odd-one-out. We define the odd-one-out in the neural network representation space analogously and measure neural networks' alignment with human similarity judgments in terms of their *odd-one-out accuracy*, i.e., the accuracy of their odd-one-out "judgments" with respect to humans', under a wide variety of settings. We confirm our findings on two independent datasets collected using the multi-arrangement task, in which humans arrange images according to their similarity Cichy et al. (2019); King et al. (2019). Based on these analyses, we draw the following conclusions:

- Scaling ImageNet models improves ImageNet accuracy, but does not consistently improve alignment of their representations with human similarity judgments. Differences in alignment across ImageNet models arise primarily from differences in objective functions and other hyperparameters rather than from differences in architecture or width/depth.

- Models trained on image/text data, or on larger, more diverse classification datasets than ImageNet, achieve substantially better alignment with humans.

- A linear transformation trained to improve odd-one-out accuracy on THINGS substantially increases the degree of alignment on held-out THINGS images as well as for two human similarity judgment datasets that used a multi-arrangement task to collect behavioral responses.

- We use a sparse Bayesian model of human mental representations (Muttenthaler et al., 2022) to partition triplets by the concept that distinguishes the odd-one-out. While food and animal-related concepts can easily be recovered from neural net representations, human alignment is weak for dimensions that depict sports-related or royal objects, especially for ImageNet models.

## 2 RELATED WORK

Most work comparing neural networks with human behavior has focused on the errors made during image classification. Although ImageNet-trained models appear to make very different errors than humans (Rajalingham et al., 2018; Geirhos et al., 2020; 2021), models trained on larger datasets than ImageNet exhibit greater error consistency (Geirhos et al., 2021). Compared to humans, ImageNet-trained models perform worse on distorted images (RichardWebster et al., 2019; Dodge & Karam, 2017; Hosseini et al., 2017; Geirhos et al., 2018) and rely more heavily on texture cues and less on object shapes (Geirhos et al., 2019; Baker et al., 2018), although reliance on texture can be mitigated through data augmentation (Geirhos et al., 2019; Hermann et al., 2020; Li et al., 2021), adversarial training (Geirhos et al., 2021), or larger datasets (Bhojanapalli et al., 2021).

Previous work has also compared human and machine semantic similarity judgments, generally using smaller sets of images and models than we explore here. Jozwik et al. (2017) measured the similarity of AlexNet and VGG-16 representations to human similarity judgments of 92 object images inferred from a multi-arrangement task. Peterson et al. (2018) compared representations of five neural networks to pairwise similarity judgments for six different sets of 120 images. Aminoff et al. (2022) found that, across 11 networks, representations of contextually associated objects (e.g., bicycles and helmets) were more similar than those of non-associated objects; similarity correlated with both human ratings and reaction times. Roads & Love (2021) collect human similarity judgments for ImageNet and evaluate triplet accuracy on these similarity judgments using 12 ImageNet networks. Most closely related to our work, Marjieh et al. (2022) measure aligment between representations of networks that process images, videos, audio, or text and the human pairwise similarity judgments of Peterson et al. (2018). They report a weak correlation between parameter count and alignment, but do not systematically examine factors that affect this relationship.

Other studies have focused on perceptual rather than semantic similarity, where the task measures perceived similarity between a reference image and a distorted version of that reference image (Ponomarenko et al., 2009; Zhang et al., 2018), rather than between distinct images as in our task. Whereas the representations best aligned with human perceptual similarity are obtained from intermediate layers of small architectures (Berardino et al., 2017; Zhang et al., 2018; Chinen et al., 2018; Kumar et al., 2022), the representations best aligned with our odd-one-out judgments are obtained at final model layers, and architecture has little impact. Jagadeesh & Gardner (2022) compared human odd-one-out judgments with similarities implied by neural network representations and

brain activity. They found that artificial and biological representations distinguish the odd one out when it differs in category, but do not distinguish natural images from synthetic scrambled images.

Our work fits into a broader literature examining relationships between in-distribution accuracy of image classification and other model quality measures, including accuracy on out-of-distribution (OOD) data and downstream accuracy when transferring the model. OOD accuracy correlates nearly linearly with accuracy on the training distribution (Recht et al., 2019; Taori et al., 2020; Miller et al., 2021), although data augmentation can improve accuracy under some shifts without improving in-distribution accuracy (Hendrycks et al., 2021). When comparing the transfer learning performance across different architectures trained with similar settings, accuracy on the pretraining task correlates well with accuracy on the transfer tasks (Kornblith et al., 2019b), but differences in regularization, training objective, and hyperparameters can affect linear transfer accuracy even when the impact on pretraining accuracy is small (Kornblith et al., 2019b; 2021; Abnar et al., 2022). In our study, we find that the training objective has a significant impact, as it does for linear transfer. However, in contrast to previous observations regarding OOD generalization and transfer, we find that better-performing architectures do not achieve greater human alignment.

# 3 METHODS

## 3.1 DATA

Our primary analyses use images and corresponding human odd-one-out triplet judgments from the THINGS dataset (Hebart et al., 2019). THINGS consists of a collection of 1,854 object categories, concrete nameable nouns in the English language that can be easily identified as a central object in a natural image, along with representative images for these categories. For presentation purposes, we have replaced the im-

Human

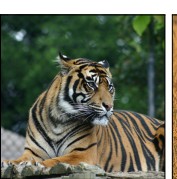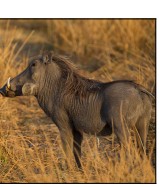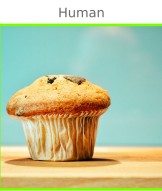

Figure 1: An example triplet from Hebart et al. (2020), where neural nets choose a different odd-one-out than a human. The images in this triplet are copyright-free images from THINGS + (Stoinski et al., 2022).

ages used in Hebart et al. (2020) with images similar in appearance that are licensed under CC0 (Stoinski et al., 2022). We additionally consider two datasets of images with human similarity judgments obtained from a multi-arrangement task (Cichy et al., 2019; King et al., 2019). We briefly describe the procedures that were used to obtain these datasets below.

**THINGS triplet task** Hebart et al. (2020) collected similarity judgments from human participants on images in THINGS in the form of responses to a *triplet task*. In this task, images from three distinct categories are presented to a participant, and the participant selects the image that is most different from the other two (or equivalently the pair that are most similar). The triplet task has been used to study properties of human mental representation for many decades (e.g., Fukuzawa et al., 1988; Robilotto & Zaidi, 2004; Hebart et al., 2020). Compared to tasks involving numerical/Likert-scale pairwise similarity judgments, the triplet task does not require different subjects to interpret the scale similarly and does not require that the degree of perceived similarity is cognitively accessible.

Hebart et al. (2020) collected 1.46 million unique responses crowdsourced from 5,301 workers. See Figure 1 for an example triplet. Some triplets offer an obvious answer to the triplet task, e.g. "cat", "dog", "candelabra", whereas others can be ambiguous, e.g. "knife", "table", "candelabra." To estimate the consistency of triplet choices among participants Hebart et al. (2020) collected 25 responses for each triplet in a randomly selected set of 1,000 triplets. From these responses, Hebart et al. (2020) determined that the maximum achievable odd-one-out accuracy is $67.22\% \pm 1.04\%$.

**Multi-arrangement task** The multi-arrangement task is another task commonly used to measure human similarity judgments (Kriegeskorte & Mur, 2012). In this task, subjects arrange images on a computer screen so that the distances between them reflect their similarities. We use multi-arrangement task data from two recent studies. Cichy et al. (2019) collected similarity judgments from 20 human participants for 118 natural images from ImageNet (Deng et al., 2009), and King et al. (2019) collected similarity judgments from 20 human participants for two natural image sets with 144 images per image set. The 144 images correspond to 48 object categories, each with three images. For simplicity, we report results based on only one of these two sets of images.

## 3.2 METRICS

**Zero-shot odd-one-out accuracy** To measure alignment between humans and neural networks on the THINGS triplet task, we examine the extent to which the odd-one-out can be identified directly from the similarities between images in models' representation spaces. Given representations $x_1$, $x_2$, and $x_3$ of the three images that comprise the triplet, we first construct a similarity matrix $S \in \mathbb{R}^{3 \times 3}$ where $S_{i,j} := x_i^\top x_j / (\|x_i\|_2 \|x_j\|_2)$, the cosine similarity between a pair of representations.[1] We identify the closest pair of images in the triplet as $\arg\max_{i,j>i} S_{i,j}$; the remaining image is the odd-one-out. We define zero-shot odd-one-out accuracy as the proportion of triplets where the odd-one-out identified in this fashion matches the human odd-one-out response. When evaluating zero-shot odd-one-out accuracy of supervised ImageNet models, we report the better of the accuracies obtained from representations of the penultimate embedding layer and logits; for self-supervised models, we use the projection head input; for image/text models, we use the representation from the joint image/text embedding space; and for the JFT-3B model, we use the penultimate layer. In Figure B.1 we show that representations obtained from earlier layers performed worse than representations from top layers, as in previous work (Montavon et al., 2011).

**Probing** In addition to measuring zero-shot odd-one-out accuracy on THINGS, we also learn a linear transformation of each neural network's representation that maximizes odd-one-out accuracy and then measure odd-one-out accuracy of the transformed representation on triplets comprising a held-out set of images. Following Alain & Bengio (2017), we refer to this procedure as linear probing. To learn the linear probe, we formulate the notion of the odd-one-out probabilistically, as in Hebart et al. (2020). Given image similarity matrix $S$ and a triplet $\{i, j, k\}$ (here the images are indexed by natural numbers), the likelihood of a particular pair, $\{a, b\} \subset \{i, j, k\}$, being most similar, and thus the remaining image being the odd-one-out, is modeled by the softmax of the object similarities,

$$p(\{a,b\}|\{i,j,k\}, S) := \exp(S_{a,b}) / \left(\exp(S_{i,j}) + \exp(S_{i,k}) + \exp(S_{j,k})\right). \quad (1)$$

We learn the linear transformation that maximizes the log-likelihood of the triplet odd-one-out judgments plus an $\ell_2$ regularization term. Specifically, given triplet responses $(\{a_s, b_s\}, \{i_s, j_s, k_s\})_{s=1}^n$ we find a square matrix $W$ yielding a similarity matrix $S_{ij} = (W x_i)^\top (W x_j)$ that optimizes

$$\underset{W}{\arg\min} \quad -\frac{1}{n} \sum_{s=1}^n \log \underbrace{p\left(\{a_s, b_s\}|\{i_s, j_s, k_s\}, S\right)}_{\text{odd-one-out prediction}} + \lambda \|W\|_2^2. \quad (2)$$

Here, we determine $\lambda$ via grid-search during $k$-fold cross-validation (CV). To obtain a minimally biased estimate of the odd-one-out accuracy of a linear probe, we partition the $m$ objects into two disjoint sets. Experimental details about the optimization process, $k$-fold CV, and how we partition the objects can be found in Appendix A.1 and Algorithm 1 respectively.

**RSA** To measure the alignment between human and neural net representation spaces on multi-arrangement datasets, following previous work, we perform representational similarity analysis (RSA; Kriegeskorte et al. (2008)) and compute correlation coefficients between neural network and human representational similarity matrices (RSMs) for the same sets of images (Kriegeskorte & Kievit, 2013; Cichy et al., 2019). We construct RSMs using a Pearson correlation kernel and measure the Spearman correlation between RSMs. We measure alignment on multi-arrangement datasets in a zero-shot setting as well as after applying the linear probe $W$ learned on THINGS.

## 3.3 MODELS

**Vision models** In our evaluation, we consider a diverse set of pretrained neural networks, including a wide variety of self-supervised and supervised models trained on ImageNet-1K and ImageNet-21K; three models trained on EcoSet (Mehrer et al., 2021), which is another natural image dataset; a "gigantic" Vision Transformer trained on the proprietary JFT-3B dataset (ViT-G/14 JFT) (Zhai et al., 2022); and image/text models CLIP (Radford et al., 2021), ALIGN (Jia et al., 2021), and BA-SIC (Pham et al., 2022). A comprehensive list of models that we analyze can be found in Table B.1. To obtain image representations, we use `thingsvision`, a Python library for extracting activations from neural nets (Muttenthaler & Hebart, 2021). We determine the ImageNet top-1 accuracy for networks not trained on ImageNet-1K by training a linear classifier on the network's penultimate layer using L-BFGS (Liu & Nocedal, 1989).

---

[1]We use cosine similarity rather than dot products because it nearly always yields similar or better zero-shot odd-one-out accuracies, as shown in Figure B.2.

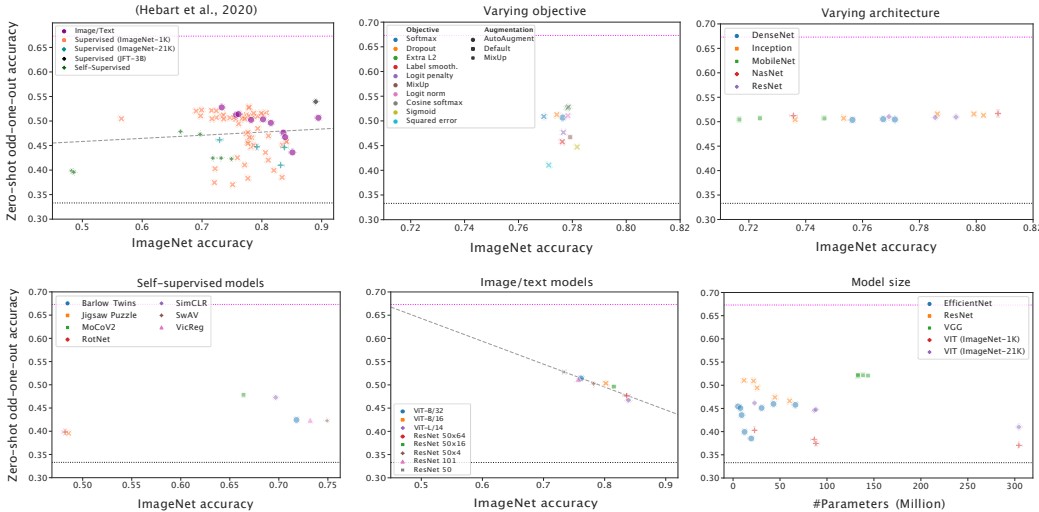

Figure 2: Zero-shot odd-one-out accuracy on THINGS only weakly correlates with ImageNet accuracy and varies with training objective but not with model architecture. **Top left**: Zero-shot accuracy as a function of ImageNet accuracy for all models. Diagonal line indicates least-squares fit. **Top center**: Models with the same architecture (ResNet-50) trained with a different objective function or different data augmentation. Since MixUp alters both inputs and targets, it is listed under both objectives and augmentations. **Top right**: Models trained with the same objective (softmax cross-entropy) but with different architectures. **Bottom left**: Performance of different SSL models. **Bottom center**: Zero-shot accuracy is negatively correlated with ImageNet accuracy for image/text models. **Bottom right**: A subset of ImageNet models with their number of parameters, colored by model family. Note that, in this subplot, models that belong to different families come from different sources and were trained with different objectives, hyperparameters, etc.; thus, models are only directly comparable within a family. In all plots, horizontal lines reflect chance-level or ceiling accuracy. See also Table B.1.

**VICE** Several of our analyses make use of human concept representations obtained by Variational Interpretable Concept Embeddings (VICE), an approximate Bayesian method for finding representations that explain human odd-one-out responses in a triplet task (Muttenthaler et al., 2022). VICE uses mean-field VI to learn a sparse representation for each image that explains the associated behavioral responses. VICE achieves an odd-one-out accuracy of $\sim 64\%$ on THINGS, which is only marginally below the ceiling accuracy of $67.22\%$ (Hebart et al., 2020). The representation dimensions obtained from VICE are highly interpretable and thus give insight into properties humans deem important for similarity judgments. We use the representation dimensions to analyze the alignment of neural network representations with human concept spaces. However, VICE is not a vision model, and can only predict odd-one-out judgments for images included in the training triplets.

## 4 EXPERIMENTS

Here, we investigate how closely the representation spaces of neural networks align with humans' concept spaces, and whether concepts can be recovered from a representation via a linear transformation. Odd-one-out accuracies are measured on THINGS, unless otherwise stated.

### 4.1 ODD-ONE-OUT VS. IMAGENET ACCURACY

We begin by comparing zero-shot odd-one-out accuracy for THINGS with ImageNet accuracy for all models in Figure 2 (top left). ImageNet accuracy generally is a good predictor for transfer learning performance (Kornblith et al., 2019b; Djolonga et al., 2021; Ericsson et al., 2021). However, while ImageNet accuracy is highly correlated with odd-one-out accuracy for a reference triplet task that uses the CIFAR-100 superclasses (see Appendix C), its correlation with accuracy on human odd-one-out judgments is very weak ($r = 0.099$). This raises the question of whether there are model, task, or data characteristics that influence human alignment.

**Architecture or objective?** We investigate odd-one-out accuracy as a function of ImageNet accuracy for models that vary in the training objective/final layer regularization, data augmentation, or architecture with all other hyperparameters fixed. Models with the same architecture (ResNet-50) trained with different objectives (Kornblith et al., 2021) yield substantially different zero-shot odd-one-out accuracies (Figure 2 top center). Conversely, models with different architectures trained

with the same objective (Kornblith et al., 2019b) achieve similar odd-one-out accuracies, although their ImageNet accuracies vary significantly (Figure 2 top right). Thus, whereas architecture does not appear to affect odd-one-out accuracy, training objective has a significant effect.

Training objective also affects which layer yields the best human alignment. For networks trained with vanilla softmax cross-entropy, the logits layer consistently yields higher zero-shot odd-one-out accuracy than the penultimate layer, but among networks trained with other objectives, the penultimate layer often provides higher odd-one-out accuracy than the logits (Figure E.2). The superiority of the logits layer of networks trained with vanilla softmax cross-entropy is specific to the odd-one-out task and RSA and does not hold for linear transfer, as we show in Appendix D.

**Self-supervised learning** Jigsaw (Noroozi & Favaro, 2016) and RotNet (Gidaris et al., 2018) show substantially worse alignment with human judgments than other SSL models (Figure 2 bottom left). This is not surprising given their poor performance on ImageNet. Jigsaw and RotNet are the only SSL models in our analysis that are non-Siamese, i.e., they were not trained by connecting two augmented views of the same image. For Siamese networks, however, ImageNet performance does not correspond to alignment with human judgments. SimCLR (Chen et al., 2020) and MoCo-v2 (He et al., 2020), both trained with a contrastive learning objective, achieve higher zero-shot odd-one-out accuracy than Barlow Twins (Zbontar et al., 2021), SwAV (Caron et al., 2020), and VICReg (Bardes et al., 2022)—of which all were trained with a non-contrastive learning objective—although their ImageNet performances are reversed. This indicates that contrasting positive against negative examples rather than using positive examples only improves alignment with human similarity judgments.

**Model capacity** Whereas one typically observes a positive correlation between model capacity and task performance in computer vision (Tan & Le, 2019; Kolesnikov et al., 2020; Zhai et al., 2022), we observe no relationship between model parameter count and odd-one-out accuracy (Figure 2 bottom right). Thus, scaling model width/depth alone appears to be ineffective at improving alignment.

## 4.2 CONSISTENCY OF RESULTS ACROSS DIFFERENT DATASETS

Although the multi-arrangement task is quite different from the triplet odd-one-out task, we observe similar results for both human similarity judgment datasets that leverage this task (see Figure 3). Again, ImageNet accuracy is not correlated with the degree of human alignment, and objective function and training data, but not architecture or model size, have a substantial impact.

## 4.3 HOW MUCH ALIGNMENT CAN A LINEAR PROBE RECOVER?

We next investigate human alignment of neural network representations after linearly transforming the representations to improve odd-one-out accuracy, as described in §3. In addition to evaluating probing odd-one-out accuracies, we perform RSA after applying the transformation matrix obtained from linear probing on the triplet odd-one-out task to a model's raw representation space. Note that the linear probe was trained exclusively on a subset of triplets from Hebart et al. (2020) (see Appendix E), without access to human responses from the two other human similarity judgments datasets (Cichy et al., 2019; King et al., 2019).

Across all three datasets, we observe that the transformation matrices obtained from linear probing substantially improve the degree of alignment with human similarity judgments. The probing odd-one-out accuracies are correlated with the zero-shot odd-one-out accuracies for both the embedding (Figure 4 left; $\rho = 0.774$) and the logit layers (Figure E.1; $\rho = 0.880$). Similarly, we observe a strong correlation between the human alignment of raw and transformed representation spaces for the embedding layer for both multi-arrangement task datasets from Cichy et al. (2019) (Figure 4 center; $\rho = 0.749$) and King et al. (2019) (Figure 4 right; $\rho = 0.519$) respectively. After applying the transformation matrices to neural nets' representations, we find that image/text models and ViT-G /14 JFT are better aligned than ImageNet or EcoSet models for all datasets and metrics.

As we discuss further in Appendix E, the relationship between probing odd-one-out accuracy and ImageNet accuracy is generally similar to the relationship between zero-shot odd-one-out accuracy and ImageNet accuracy. The same holds for the relationship between Spearman's $\rho$ and ImageNet accuracy and Spearman's $\rho$ (+ transform) and ImageNet accuracy. The correlation between ImageNet accuracy and probing odd-one-out accuracy remains weak ($r = 0.222$). Probing reduces the variance in odd-one-out accuracy or Spearman's $\rho$ among networks trained with different loss functions, self-supervised learning methods, and image/text models, yet we still fail to see improvements in probing accuracy with better-performing architectures or larger model capacities. However,

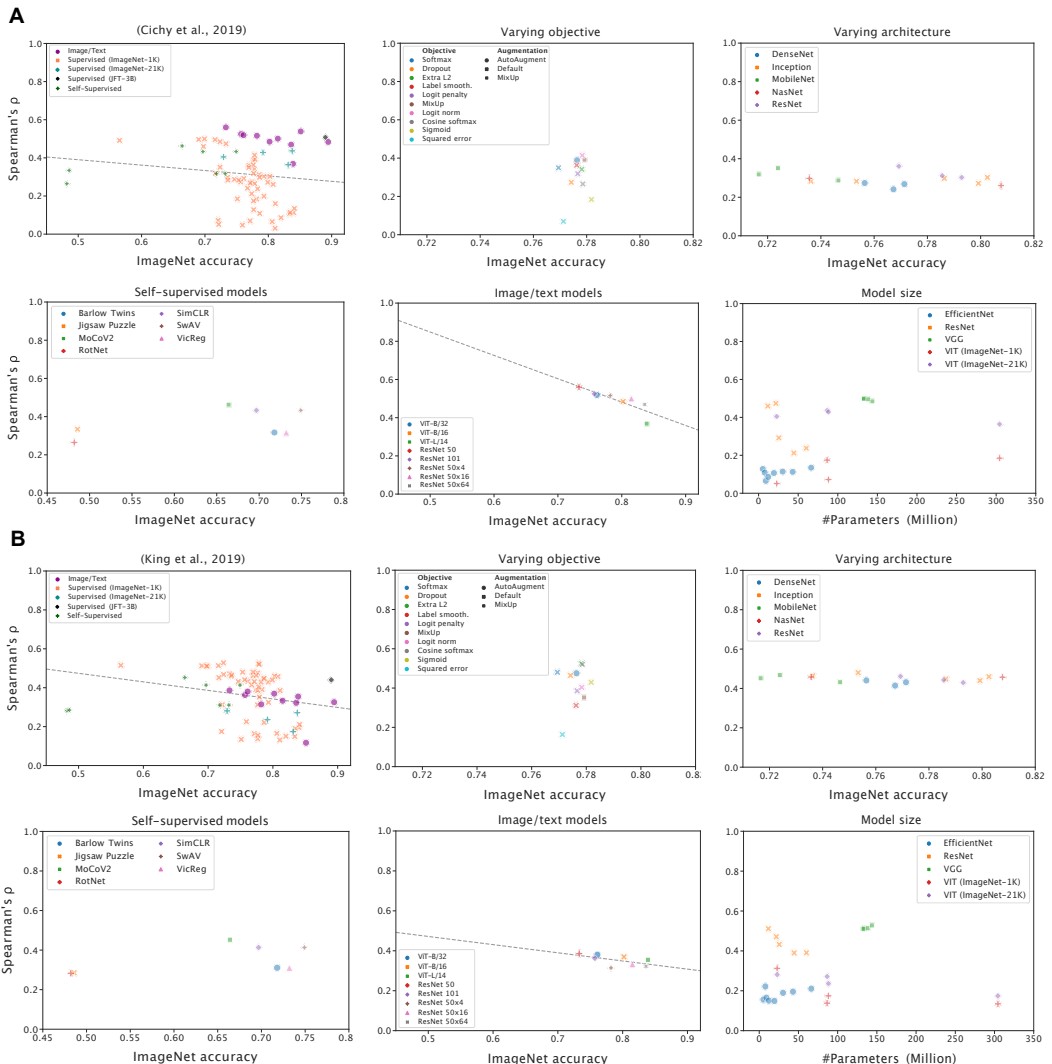

Figure 3: Spearman correlation between human and neural network representational similarity matrices is not correlated with ImageNet accuracy for ImageNet models and is negatively correlated for image/text models. Alignment varies with training objective but not with model architecture or number of parameters for both similarity judgment datasets (Cichy et al., 2019; King et al., 2019). See caption of Figure 2 for further description of panels. Diagonal lines indicate least-squares fits.

whereas image/text models exhibit a negative correlation between ImageNet accuracy and zero-shot odd-one-out accuracy is negative in Figures 2 and 3, the correlation between ImageNet accuracy and probing odd-one-out accuracy is small but positive.

Interestingly, for EcoSet models, transformation matrices do not improve alignment as much as they do for architecturally identical ImageNet models. Although one goal of EcoSet was to provide data that yields better alignment with human perception than ImageNet (Mehrer et al., 2021), we find that models trained on EcoSet are less aligned with human similarity judgments than ImageNet models.

## 4.4 HOW WELL DO PRETRAINED NEURAL NETS REPRESENT HUMAN CONCEPTS?

Below, we examine zero-shot and linear probing odd-one-out accuracies for individual human concepts. To investigate how well neural nets represent these concepts, we filter the original dataset $\mathcal{D}$ to produce a new dataset $\mathcal{D}^*$ containing only triplets correctly predicted by VICE. Thus, the best attainable odd-one-out accuracy for any model is 1 as opposed to the upper-bound of $0.6722$ for the full data. We further partition $\mathcal{D}^*$ into 45 subsets according to the 45 VICE dimensions, $\mathcal{D}_1^*, \ldots, \mathcal{D}_{45}^*$. A triplet belongs to $\mathcal{D}_j^*$ when the sum of the VICE representations for the two most similar objects in the triplet, $\boldsymbol{x}_a, \boldsymbol{x}_b$, attains its maximum in dimension $j$, $j = \arg\max_{j'} x_{a,j'} + x_{b,j'}$.

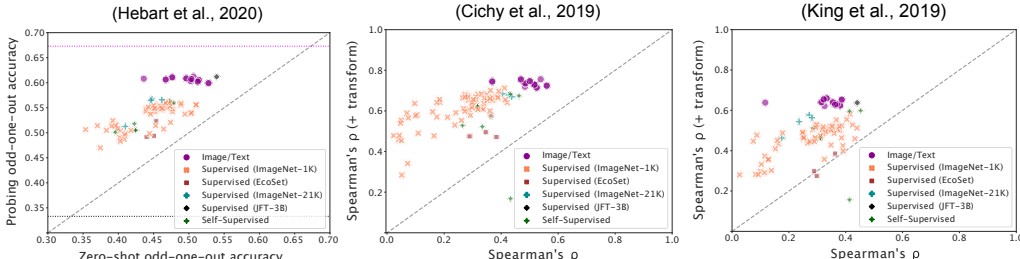

Figure 4: Left panel: Zero-shot and probing odd-one-out accuracies for the embedding layer of all neural nets. Right panels: Spearman rank correlation coefficients with and without applying the transformation matrix obtained from linear probing to a model's raw representation space. Dashed lines indicate $y = x$.

### 4.4.1 HUMAN ALIGNMENT IS CONCEPT-SPECIFIC

In Figure 5, we show zero-shot and linear probing odd-one-out accuracies for a subset of three of the 45 VICE dimensions for a large subset of the models listed in Table B.1. Zero-shot and probing odd-one-out accuracies for a larger set of dimensions can be found in Appendix J. Since the dimensions found for THINGS are similar both in visual appearance and in their number between Muttenthaler et al. (2022) and Hebart et al. (2020), we infer a labeling of the human dimensions from Hebart et al. (2020) who have evidenced the interpretability of these dimensions through human experiments.

Although models trained on large datasets — image/text models and ViT-G/14 JFT — generally show a higher zero-shot odd-one-out accuracy compared to self-supervised models or models trained on ImageNet, the ordering of models is not entirely consistent across concepts. For dimension 10 (vehicles), ResNets trained with a cosine softmax objective were the best zero-shot models, whereas image/text models were among the worst. For dimension 4, an animal-related concept, models pretrained on ImageNet clearly show the worst performance, whereas this concept is well represented in image/text models. Differences in the representation of the animal concept across models are additionally corroborated by the t-SNE visualizations in Appendix H.

Linear probing yields more consistent patterns than zero-shot performance. For almost every human concept, image/text models and ViT-G/14 JFT have the highest per-concept odd-one-out accuracies, whereas AlexNet and EfficientNets have the lowest. This difference is particularly apparent for dimension 17, which summarizes sports-related objects. For this dimension, image/text models and ViT-G/14 JFT perform much better than all remaining models. As shown in Appendix G, even for triplets where VICE predicts that human odd-one-out judgments are very consistent, ImageNet models make a substantial number of errors. By contrast, image/text models and ViT-G/14 JFT achieve a near-zero zero-shot odd-one-out error for these triplets.

### 4.4.2 CAN HUMAN CONCEPTS BE RECOVERED VIA LINEAR REGRESSION?

To further understand the extent to which human concepts can be recovered from neural networks' representation spaces, we perform $\ell_2$-regularized linear regression to examine models' ability to predict VICE dimensions. The results from this analysis – which we present in detail in Appendix F – corroborate the findings from §4.3: models trained on image/text data and ViT-G/14 JFT consistently provide the best fit for VICE dimensions, while AlexNet and EfficientNets show the poorest regression performance. We compare odd-one-out accuracies after linear probing and regression respectively. The two performance measures are highly correlated for the embedding ($r = 0.982$) and logit ($r = 0.972$; see Figure F.3) layers, which additionally supports our observations from linear probing. Furthermore, we see that the leading VICE dimensions, which are the most important for explaining human triplet responses, could be fitted with an $R^2$ score of $> 0.7$ for most of the models – the quality of the regression fit declines with the importance of a dimension (see Figure F.4).

## 5 DISCUSSION

In this work, we evaluated the alignment of neural network representations with human concepts spaces through performance in an odd-one-out task and by performing representational similarity analysis. Before discussing our findings we will address limitations of our work. One limitation is

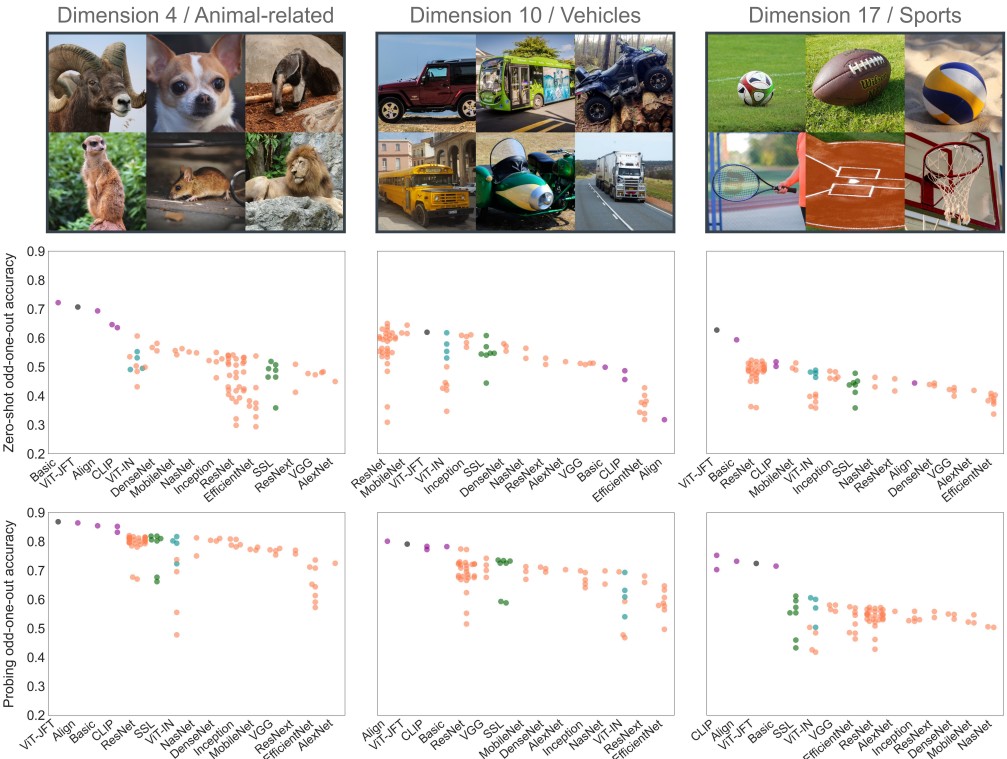

Figure 5: Zero-shot and linear probing odd-one-out accuracies differ across VICE concepts. Results are shown for the embedding layer of all models for three of the 45 VICE dimensions. See Appendix J for additional dimensions. Color-coding is determined by training data/objective. Violet: Image/Text. Green: Self-supervised. Orange: Supervised (ImageNet-1K). Cyan: Supervised (ImageNet-21K). **Black**: Supervised (JFT-3B).

that we did not consider non-linear transformations. It is possible that simple non-linear transformations could provide better alignment for the networks we investigate. We plan to investigate such transformations further in future work. Another limitation stems from our use of pretrained models for our experiments, since they have been trained with a variety of objectives and regularization strategies. We have mitigated this by comparing controlled subsets of models in Figure 2.

Nevertheless, we can draw the following conclusions from our analyses. First, scaling ImageNet models does not lead to better alignment of their representations with human similarity judgments. Differences in human alignment across ImageNet models are mainly attributable to the objective function with which a model was trained, whereas architecture and model capacity are both insignificant. Second, models trained on image/text or more diverse data achieve much better alignment than ImageNet models. Albeit not consistent for zero-shot odd-one-out accuracy, this is clear in both linear probing and regression results. These conclusions hold for all three datasets we have investigated, indicating that they are true properties of human/machine alignment rather than idiosyncrasies of the task. Finally, good representations of concepts that are important to human similarity judgments can be recovered from neural network representation spaces. However, representations of less important concepts, such as sports and royal objects, are more difficult to recover.

How can we train neural networks that achieve better alignment with human concept spaces? Although our results indicate that large, diverse datasets improve alignment, all image/text and JFT models we investigate all attain probing accuracies of 60-61.5%. By contrast, VICE representations achieve 64%, and a Bayes-optimal classifier achieves 67%. Since our image/text models are trained on datasets of varying sizes (400M to 6.6B images) but achieve similar alignment, we suspect that further scaling of dataset size is unlikely to close this gap. To obtain substantial improvements, it may be necessary to incorporate additional forms of supervision when training the representation itself. Benefits of improving human/machine alignment may extend beyond accuracy on our triplet task, to transfer and retrieval tasks where it is important to capture human notions of similarity.

ACKNOWLEDGEMENTS

LM, LL, and RV acknowledge support from the Federal Ministry of Education and Research (BMBF) for the Berlin Institute for the Foundations of Learning and Data (BIFOLD) (01IS18037A). This work is in part supported by Google. We thank Klaus-Robert Müller, Robert Geirhos, Katherine Hermann, and Andrew Lampinen for their helpful comments on the manuscript.

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

# A EXPERIMENTAL DETAILS

## A.1 LINEAR PROBING

**Initialization** We initialized the transformation matrix $\boldsymbol{W} \in \mathbb{R}^{p \times p}$ used in Eq. 2 with values from a tight Gaussian centered around 0, such that $\boldsymbol{W} \sim \mathcal{N}(0, 10^{-3}\boldsymbol{I})$ at the beginning of the optimization process.

**Training** We optimized the transformation matrix $\boldsymbol{W}$ via gradient descent, using Adam (Kingma & Ba, 2015) with a learning rate of $\eta = 0.001$. We performed a grid-search over the learning rate $\eta$, where $\eta \in \{0.0001, 0.001, 0.01\}$ and found $0.001$ to work best for all models in Table B.1. We trained the linear probe for a maximum of 100 epochs and stopped the optimization process early whenever the generalization performance did not change by a factor of $0.0001$ for $T = 10$ epochs. For most of our evaluation and linear probing experiments, we use PyTorch (Paszke et al., 2019).

**Cross-validation** To obtain a minimally biased estimate of the odd-one-out accuracy of a linear probe, we performed $k$-fold CV over objects rather than triplets. We partitioned the $m$ objects into two disjoint sets for train and test triplets. Algorithm 1 demonstrates how object partitioning was performed for each of the $k$ folds.

---

**Algorithm 1** Algorithm for object partitioning during $k$-fold CV

---

**Input:** $(\mathcal{D}, m)$         ▷ Here, $\mathcal{D} := (\{a_s, b_s\}, \{i_s, j_s, k_s\})_{s=1}^n$ and $m$ is the number of objects
  $[m] = \{1, \ldots, m\}$                                ▷ $|[m]| = m$
  $\mathbb{O}_{\text{train}} \sim \mathcal{U}([m])$    ▷ Sample a number of train objects uniformly at random without replacement
  $\mathbb{O}_{\text{test}} := [m] \setminus \mathbb{O}_{\text{train}}$                           ▷ Test objects are the remaining objects
  $\mathcal{D}_{\text{train}} := \{\}$                          ▷ Initialize an empty set for the train triplets
  $\mathcal{D}_{\text{test}} := \{\}$                           ▷ Initialize an empty set for the test triplets
  **for** $s \in \{1, \ldots, n\}$ **do**
      assignments $\triangleq$ list( )   ▷ For each triplet initialize an empty list to control object assignments
      **for** $x \in \{i_s, j_s, k_s\}$ **do**
         **if** $(x \in \mathbb{O}_{\text{train}})$ **then**
            assignment $\triangleq$ "train"
         **else**
            assignment $\triangleq$ "test"
         **end if**
         assignments $\leftarrow$ assignment       ▷ Append current assignment to the list of assignments
      **end for**
      **if** $(\text{len}(\text{set}(\text{assignments})) \neq 1)$ **then**
         **continue**    ▷ If not all objects in a triplet belong to the same set of objects, discard triplet
      **else**
         assignment $\triangleq$ pop(set(assignments))       ▷ Get object set assignment of current triplet
         **if** (assignment **is** "train") **then**
            $\mathcal{D}_{\text{train}} := \mathcal{D}_{\text{train}} \cup \mathcal{D}_s$                 ▷ Assign current triplet to the train set
         **else**
            $\mathcal{D}_{\text{test}} := \mathcal{D}_{\text{test}} \cup \mathcal{D}_s$                 ▷ Assign current triplet to the test set
         **end if**
      **end if**
  **end for**
**Output:** $(\mathcal{D}_{\text{train}}, \mathcal{D}_{\text{test}})$               ▷ Return both train and test triplet sets

---

Note that the number of train objects that are sampled uniformly at random without replacement from the set of all objects is dependent on $k$. We performed a grid-search search over $k$, where $k \in \{2, 3, 4, 5\}$, and observed that 3-fold and 4-fold CV lead to the best linear probing results. Since the objects in the train and test triplets are not allowed to overlap, loss of data was inevitable (see Algorithm 1). One can easily see that minimizing the loss of triplet data comes at the cost of disproportionally decreasing the size of the test set. We decided to proceed with 3-fold CV in our final experiments since using $2/3$ of the objects for training and $1/3$ for testing resulted in a proportionally larger test set than using $3/4$ for training and $1/4$ for testing ($\sim 433$k train and $\sim 54$k test triplets for 3-fold CV vs. $\sim 616$k train and $\sim 23$k test triplets for 4-fold CV). In

general, the larger a test set, the more accurate the estimate of a model's generalization performance. To find the optimal strength of the $\ell_2$ regularization for each linear probe, we performed a grid-search over $\lambda$ for each $k$ value individually. The optimal $\lambda$ varied between models, where $\lambda \in \{0.0001, 0.001, 0.01, 0.1, 1\}$.

## A.2 TEMPERATURE SCALING

It is widely known that classifiers trained to minimize cross-entropy tend to be overconfident in their predictions (Szegedy et al., 2016; Guo et al., 2017; Roelofs et al., 2022), which is in stark contrast to the high-entropy predictions of VICE. For this purpose, we performed temperature scaling (Guo et al., 2017) on the model outputs for THINGS and searched over the scaling parameter $\tau$ for each model. In particular, we considered temperature-scaled predictions

$$p(\{a,b\}|\{i,j,k\},\tau \boldsymbol{S}) = \frac{\exp(\tau S_{a,b})}{\exp(\tau S_{i,j}) + \exp(\tau S_{i,k}) + \exp(\tau S_{j,k})},$$

where we multiply $\boldsymbol{S}$ in Eq. 1 by a constant $\tau > 0$ and $S_{i,j}$ is the inner product of the model representations for images $i$ and $j$, i.e. the zero-shot similarities. There are several conceivable criteria that could be minimized to find the optimal scaling parameter $\tau$ from a set of candidates. For our analyses, we considered the following,

- Average Jensen-Shannon (JS) distance between model zero-shot probabilities and VICE probabilities over all triplets
- Average Kullback-Leibler divergence (KLD) between model zero-shot probabilities and VICE probabilities over all triplets
- Expected Calibration Error (ECE) (Guo et al., 2017)

The ECE is defined as follows. Let $\mathcal{D} = (\{a_s, b_s\}, \{i_s, j_s, k_s\})_{s=1}^n$ be the set of triplets and human responses from Hebart et al. (2020). For a given triplet $\{i, j, k\}$ and similarity matrix $\boldsymbol{S}$ we define confidence as

$$\operatorname{conf}(\{i,j,k\}, \boldsymbol{S}) := \max_{\{a,b\} \subset \{i,j,k\}} p(\{a,b\} \mid \{i,j,k\}, \boldsymbol{S}).$$

This corresponds to the expected accuracy of the Bayes classifier for that triplet according to the probability model from $\boldsymbol{S}$ with Eq. 1. We define $B_m(\boldsymbol{S})$ to be those training triplets where

$$\operatorname{conf}(\{i_s, j_s, k_s\}, \boldsymbol{S}) \in \left[\frac{m-1}{10}, \frac{m}{10}\right].$$

For a similarity matrix, $\boldsymbol{S}$, and a set of triplets with responses, $\mathcal{D}' \subset \mathcal{D}$, we define $\operatorname{acc}(\mathcal{D}', \boldsymbol{S})$ to be the portion of triplets in $\mathcal{D}'$ correctly classified according to the highest similarity according to $\boldsymbol{S}$. Finally for a set of triplets $\mathcal{D}' \subset \mathcal{D}$ and similarity matrix $\boldsymbol{S}$ we define $\operatorname{conf}(\mathcal{D}')$ to be the average confidence over that set (triplet responses are simply ignored). The ECE is defined as

$$\operatorname{ECE}(\tau, \boldsymbol{S}) = \sum_{m=1}^{10} \frac{|B_m(\tau \boldsymbol{S})|}{n} |\operatorname{acc}(B_m(\tau \boldsymbol{S})) - \operatorname{conf}(B_m(\tau \boldsymbol{S}))|.$$

Intuitively, the ECE is low if for each subset, $B_m(\tau \boldsymbol{S})$, the model's accuracy and its confidence are near each other. A model will be well-calibrated if its confidence in predicting the odd-one-out in a triplet corresponds to the probability that this prediction is correct.

Of the three considered criteria, ECE resulted in the clearest optima when varying $\tau$, whereas KLD plateaued with increasing $\tau$ and JS distance was numerically unstable, most probably because the model output probabilities were near zero for some pairs, which may result in very large JS distance. For all models, we performed a grid-search over $\tau \in \{1 \cdot 10^0, 7.5 \cdot 10^{-1}, 5 \cdot 10^{-1}, 2.5 \cdot 10^{-1}, 1 \cdot 10^{-1}, 7.5 \cdot 10^{-2}, 5 \cdot 10^{-2}, 2.5 \cdot 10^{-2}, 1 \cdot 10^{-2}, 7.5 \cdot 10^{-3}, 5 \cdot 10^{-3}, 2.5 \cdot 10^{-3}, 1 \cdot 10^{-3}, 5 \cdot 10^{-4}, 1 \cdot 10^{-4}, 5 \cdot 10^{-5}, 1 \cdot 10^{-5}\}$.

## A.3 LINEAR REGRESSION

**Cross-validation** We used ridge regression, that is $\ell_2$-regularized linear regression, to find the transformation matrix $\boldsymbol{A}_{j,:}$ and bias $b_j$ that result in the best fit. We employed nested $k$-fold CV for each of the $d$ VICE dimensions. For the outer CV we performed a grid-search over $k$, where $k \in \{2, 3, 4, 5\}$, similarly to how $k$-fold CV was performed for linear probing (see Appendix A.1). For our final experiments, we used 5-fold CV to obtain a minimally biased estimate for the $R^2$ score

of the regression fit. For the inner CV, we leveraged leave-one-out CV to determine the optimal $\alpha$ for Eq. 3 using `RidgeCV` from Pedregosa et al. (2011). We performed a grid search over $\alpha$, where $\alpha \in \{0.01, 0.1, 1, 10, 100, 1000, 10000, 100000, 1000000\}$.

## B  MODELS

First, we evaluate supervised models trained on ImageNet (Russakovsky et al., 2015), such as AlexNet (Krizhevsky, 2014), various VGGs (Simonyan & Zisserman, 2015), ResNets (He et al., 2016), EfficientNets (Tan & Le, 2019), ResNext models (Xie et al., 2017), and Vision Transformers (ViTs) trained on ImageNet-1K (Dosovitskiy et al., 2021) or ImageNet-21K (Steiner et al., 2022) respectively. Second, we analyze recent state-of-the-art models trained on image/text data, CLIP-RN & CLIP-ViT (Radford et al., 2021), ALIGN (Jia et al., 2021) and BASIC-L (Pham et al., 2022). Third, we evaluate self-supervised models that were trained with a contrastive learning objective such as SimCLR (Chen et al., 2020) and MoCo-v2 (He et al., 2020), recent SSL models that were trained with a non-contrastive learning objective (no negative examples), BarlowTwins (Zbontar et al., 2021), SwAV (Caron et al., 2020), and VICReg (Bardes et al., 2022), as well as earlier SSL, non-Siamese models, Jigsaw (Noroozi & Favaro, 2016), and RotNet (Gidaris et al., 2018). All self-supervised models have a ResNet-50 architecture. For SimCLR, MoCo-v2, Jigsaw and RotNet, we leverage model weights from the VISSL library (Goyal et al., 2021). For the other models we use weights from their official GitHub repositories. Last, we evaluate the largest available ViT (Zhai et al., 2022), trained on the proprietary JFT-3B image classification dataset, which consists of approximately three billion images belonging to approximately 30,000 classes (Zhai et al., 2022). See Table B.1 for further details regarding the models evaluated.

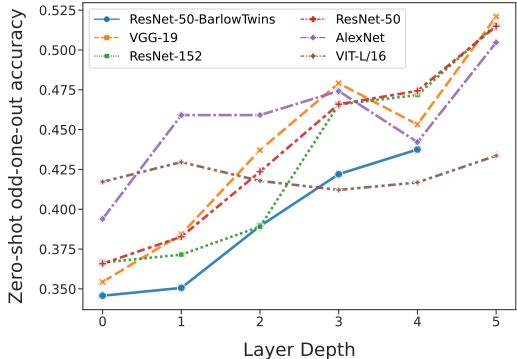

Figure B.1: Zero-shot odd-one-out accuracy for different layers for a subset of selected models.

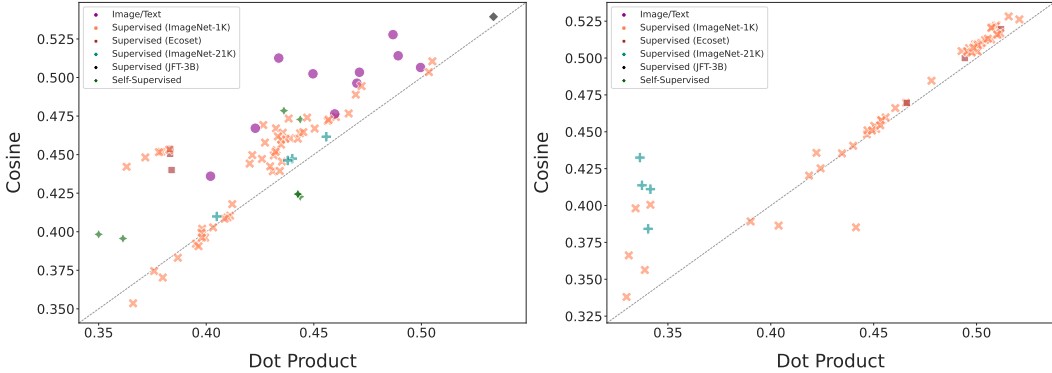

Figure B.2: Zero-shot odd-one-out accuracy using the cosine similarity nearly always is better than using the dot product as a similarity measure.

| Model | Source | Architecture | Dataset | Objective | ImageNet | Zero-Shot | Probing |
|---|---|---|---|---|---|---|---|
| AlexNet | (Krizhevsky, 2014) | AlexNet | ImageNet-1K | Supervised (softmax) | 56.52% | 50.47% | 53.84% |
| AlexNet | (Muttenthaler & Hebart, 2021) | AlexNet | Ecoset | Supervised (softmax) | - | 50.00% | 51.30% |
| ALIGN | (Jia et al., 2021) | EfficientNet | ALIGN dataset | Image/Text (contr.) | 85.11% | 43.60% | 60.81% |
| Basic-L | (Pham et al., 2022) | CoAtNet | ALIGN + JFT-5B | Image/Text (contr.) | 89.45% | 50.65% | 61.24% |
| CLIP RN101 | (Radford et al., 2021) | ResNet | CLIP dataset | Image/Text (contr.) | 75.70% | 51.26% | 60.22% |
| CLIP RN50 | (Radford et al., 2021) | ResNet | CLIP dataset | Image/Text (contr.) | 73.30% | 52.78% | 59.92% |
| CLIP RN50x16 | (Radford et al., 2021) | ResNet | CLIP dataset | Image/Text (contr.) | 81.50% | 49.63% | 60.86% |
| CLIP RN50x4 | (Radford et al., 2021) | ResNet | CLIP dataset | Image/Text (contr.) | 78.20% | 50.24% | 60.38% |
| CLIP RN50x64 | (Radford et al., 2021) | ResNet | CLIP dataset | Image/Text (contr.) | 83.60% | 47.64% | 61.07% |
| CLIP ViT-B/16 | (Radford et al., 2021) | ViT | CLIP dataset | Image/Text (contr.) | 80.20% | 50.34% | 60.72% |
| CLIP ViT-B/32 | (Radford et al., 2021) | ViT | CLIP dataset | Image/Text (contr.) | 76.10% | 51.41% | 60.54% |
| CLIP ViT-L/14 | (Radford et al., 2021) | ViT | CLIP dataset | Image/Text (contr.) | 83.90% | 46.71% | 60.64% |
| DenseNet-121 | (Kornblith et al., 2019b) | DenseNet | ImageNet-1K | Supervised (softmax) | 75.64% | 50.37% | 55.18% |
| DenseNet-169 | (Kornblith et al., 2019b) | DenseNet | ImageNet-1K | Supervised (softmax) | 76.73% | 50.52% | 55.36% |
| DenseNet-201 | (Kornblith et al., 2019b) | DenseNet | ImageNet-1K | Supervised (softmax) | 77.14% | 50.45% | 55.37% |
| EfficientNet B0 | (Tan & Le, 2019) | EfficientNet | ImageNet-1K | Supervised (softmax) | 77.69% | 45.42% | 50.82% |
| EfficientNet B1 | (Tan & Le, 2019) | EfficientNet | ImageNet-1K | Supervised (softmax) | 79.84% | 45.08% | 51.30% |
| EfficientNet B2 | (Tan & Le, 2019) | EfficientNet | ImageNet-1K | Supervised (softmax) | 80.61% | 43.23% | 49.33% |
| EfficientNet B3 | (Tan & Le, 2019) | EfficientNet | ImageNet-1K | Supervised (softmax) | 82.01% | 39.94% | 50.79% |
| EfficientNet B4 | (Tan & Le, 2019) | EfficientNet | ImageNet-1K | Supervised (softmax) | 83.38% | 38.52% | 50.65% |
| EfficientNet B5 | (Tan & Le, 2019) | EfficientNet | ImageNet-1K | Supervised (softmax) | 83.44% | 45.10% | 51.47% |
| EfficientNet B6 | (Tan & Le, 2019) | EfficientNet | ImageNet-1K | Supervised (softmax) | 84.01% | 45.97% | 51.56% |
| EfficientNet B7 | (Tan & Le, 2019) | EfficientNet | ImageNet-1K | Supervised (softmax) | 84.12% | 45.77% | 52.41% |
| Inception-RN V2 | (Kornblith et al., 2019b) | Inception | ImageNet-1K | Supervised (softmax) | 80.26% | 51.31% | 55.78% |
| Inception-V1 | (Kornblith et al., 2019b) | Inception | ImageNet-1K | Supervised (softmax) | 73.63% | 50.43% | 54.97% |
| Inception-V2 | (Kornblith et al., 2019b) | Inception | ImageNet-1K | Supervised (softmax) | 75.34% | 50.70% | 54.97% |
| Inception-V3 | (Kornblith et al., 2019b) | Inception | ImageNet-1K | Supervised (softmax) | 78.64% | 51.59% | 55.84% |
| Inception-V4 | (Kornblith et al., 2019b) | Inception | ImageNet-1K | Supervised (softmax) | 79.92% | 51.58% | 55.47% |
| MobileNet-V1 | (Kornblith et al., 2019b) | MobileNet | ImageNet-1K | Supervised (softmax) | 72.39% | 50.70% | 54.98% |
| MobileNet-V2 | (Kornblith et al., 2019b) | MobileNet | ImageNet-1K | Supervised (softmax) | 71.67% | 50.45% | 55.17% |
| MobileNet-V2 (1.4) | (Kornblith et al., 2019b) | MobileNet | ImageNet-1K | Supervised (softmax) | 74.66% | 50.67% | 55.11% |
| NASNet-L | (Kornblith et al., 2019b) | NASNet | ImageNet-1K | Supervised (softmax) | 80.77% | 51.68% | 55.78% |
| NASNet-Mobile | (Kornblith et al., 2019b) | NASNet | ImageNet-1K | Supervised (softmax) | 73.57% | 51.23% | 55.48% |
| RN-50-BarlowTwins | (Zbontar et al., 2021) | ResNet | ImageNet-1K | Self-sup. (non-contr.) | 71.80% | 42.44% | 50.50% |
| RN-50-Jigsaw | (Goyal et al., 2021) | ResNet | ImageNet-1K | Self-sup. (non-Siamese) | 48.57% | 39.56% | 50.11% |
| RN-50-MoCo-v2 | (Goyal et al., 2021) | ResNet | ImageNet-1K | Self-sup. (contr.) | 66.40% | 47.85% | 55.94% |
| RN-50-RotNet | (Goyal et al., 2021) | ResNet | ImageNet-1K | Self-sup. (non-Siamese) | 54.93% | 39.83% | 50.82% |
| RN-50-SimCLR | (Goyal et al., 2021) | ResNet | ImageNet-1K | Self-sup. (contr.) | 69.68% | 47.28% | 56.37% |
| RN-50-SWAV | (Caron et al., 2020) | ResNet | ImageNet-1K | Self-sup. (non-contr.) | 74.92% | 42.27% | 51.79% |
| RN-50-VICReg | (Bardes et al., 2022) | ResNet | ImageNet-1K | Self-sup. (non-contr.) | 73.20% | 42.44% | 50.50% |
| RN-18 | (He et al., 2016) | ResNet | ImageNet-1K | Supervised (softmax) | 69.76% | 51.05% | 54.97% |
| RN-34 | (He et al., 2016) | ResNet | ImageNet-1K | Supervised (softmax) | 73.31% | 50.93% | 55.30% |
| RN-50 | (He et al., 2016) | ResNet | ImageNet-1K | Supervised (softmax) | 80.86% | 49.44% | 53.72% |
| RN-101 | (He et al., 2016) | ResNet | ImageNet-1K | Supervised (softmax) | 81.89% | 47.40% | 52.06% |
| RN-152 | (He et al., 2016) | ResNet | ImageNet-1K | Supervised (softmax) | 82.28% | 46.61% | 50.74% |
| RN-101 | (Kornblith et al., 2019b) | ResNet | ImageNet-1K | Supervised (softmax) | 78.56% | 50.86% | 55.95% |
| RN-152 | (Kornblith et al., 2019b) | ResNet | ImageNet-1K | Supervised (softmax) | 79.29% | 50.95% | 56.04% |
| RN-50 | (Kornblith et al., 2019b) | ResNet | ImageNet-1K | Supervised (softmax) | 76.93% | 51.02% | 56.05% |
| RN-50 (dropout) | (Kornblith et al., 2021) | ResNet | ImageNet-1K | Supervised (softmax+) | 77.42% | 51.26% | 55.40% |
| RN-50 (extra WD) | (Kornblith et al., 2021) | ResNet | ImageNet-1K | Supervised (softmax+) | 77.82% | 52.62% | 56.16% |
| RN-50 (label smoothing) | (Kornblith et al., 2021) | ResNet | ImageNet-1K | Supervised (softmax+) | 77.63% | 45.78% | 55.52% |
| RN-50 (logit penality) | (Kornblith et al., 2021) | ResNet | ImageNet-1K | Supervised (softmax+) | 77.67% | 47.67% | 54.21% |
| RN-50 (mixup) | (Kornblith et al., 2021) | ResNet | ImageNet-1K | Supervised (softmax+) | 77.92% | 46.70% | 56.29% |
| RN-50 (AutoAugment) | (Kornblith et al., 2021) | ResNet | ImageNet-1K | Supervised (softmax) | 77.64% | 50.67% | 56.10% |
| RN-50 (logit norm) | (Kornblith et al., 2021) | ResNet | ImageNet-1K | Supervised (softmax+) | 77.83% | 51.05% | 55.63% |
| RN-50 (cosine softmax) | (Kornblith et al., 2021) | ResNet | ImageNet-1K | Supervised (softmax+) | 77.86% | 52.82% | 56.73% |
| RN-50 (sigmoid) | (Kornblith et al., 2021) | ResNet | ImageNet-1K | Supervised (sigmoid) | 78.18% | 44.72% | 55.34% |
| RN-50 (softmax) | (Kornblith et al., 2021) | ResNet | ImageNet-1K | Supervised (softmax) | 76.94% | 50.89% | 55.97% |
| RN-50 (squared error) | (Kornblith et al., 2021) | ResNet | ImageNet-1K | Supervised (sq. error) | 77.13% | 41.04% | 49.87% |
| RN-50 | (Muttenthaler & Hebart, 2021) | ResNet | Ecoset | Supervised (softmax) | - | 46.96% | 50.63% |
| ResNeXt-50 32x4d | (Xie et al., 2017) | ResNeXt | ImageNet-1K | Supervised (softmax) | 81.20% | 46.97% | 51.44% |
| ResNeXt-101 32x8d | (Xie et al., 2017) | ResNeXt | ImageNet-1K | Supervised (softmax) | 81.89% | 48.46% | 50.82% |
| VGG-11 | (Simonyan & Zisserman, 2015) | VGG | ImageNet-1K | Supervised (softmax) | 69.02% | 52.04% | 55.91% |
| VGG-13 | (Simonyan & Zisserman, 2015) | VGG | ImageNet-1K | Supervised (softmax) | 69.93% | 52.24% | 55.84% |
| VGG-16 | (Simonyan & Zisserman, 2015) | VGG | ImageNet-1K | Supervised (softmax) | 71.59% | 52.09% | 55.86% |
| VGG-19 | (Simonyan & Zisserman, 2015) | VGG | ImageNet-1K | Supervised (softmax) | 72.38% | 52.09% | 55.86% |
| VGG-16 | (Muttenthaler & Hebart, 2021) | VGG | Ecoset | Supervised (softmax) | - | 51.96% | 53.58% |
| ViT-B/16 I1K | (Steiner et al., 2022) | ViT | ImageNet-1K | Supervised (sigmoid) | 77.66% | 38.31% | 50.48% |
| ViT-B/16 I21K | (Steiner et al., 2022) | ViT | ImageNet-21K | Supervised (sigmoid) | 83.77% | 44.62% | 56.49% |
| ViT-B/32 I1K | (Steiner et al., 2022) | ViT | ImageNet-1K | Supervised (sigmoid) | 72.08% | 37.45% | 46.99% |
| ViT-B/32 I21K | (Steiner et al., 2022) | ViT | ImageNet-21K | Supervised (sigmoid) | 79.16% | 44.74% | 56.78% |
| ViT-L/16 I1K | (Steiner et al., 2022) | ViT | ImageNet-1K | Supervised (sigmoid) | 75.11% | 37.03% | 51.42% |
| ViT-L/16 I21K | (Steiner et al., 2022) | ViT | ImageNet-21K | Supervised (sigmoid) | 83.13% | 40.99% | 51.27% |
| ViT-S/32 I1K | (Steiner et al., 2022) | ViT | ImageNet-1K | Supervised (sigmoid) | 72.18% | 40.28% | 48.31% |
| ViT-S/32 I21K | (Steiner et al., 2022) | ViT | ImageNet-21K | Supervised (sigmoid) | 72.93% | 46.16% | 56.60% |
| ViT-G/14 JFT | (Zhai et al., 2022) | ViT | JFT-3B | Supervised (sigmoid) | 89.01% | 53.94% | 61.18% |
| ViT-B-16 | (Dosovitskiy et al., 2021) | ViT | ImageNet-1K | Supervised (softmax) | 81.07% | 42.02% | 47.89% |
| ViT-B-32 | (Dosovitskiy et al., 2021) | ViT | ImageNet-1K | Supervised (softmax) | 75.91% | 42.52% | 48.69% |

Table B.1: Pretrained neural networks that we considered in our analyses. "RN" = ResNet, "Self-sup." = self-supervised, "contr." = contrastive. "RN-50 (extra WD)" is a ResNet-50 with higher weight decay on the final network layer. The "ImageNet" column contains the accuracy on the ImageNet dataset. The "Zero-Shot" column contains the THINGS zero-shot odd-one-out accuracy of the better of the embedding and logits layer. The "Probing" column contains the THINGS probing odd-one-out accuracy of the embedding layer.

Figure B.1 shows the odd-one-out accuracy as a function of layer depth in a neural network for a few different network architectures. Later layers generally perform better which is why we performed our analyses exclusively for the logits or penultimate/embedding layers of the models in Table B.1. Figure B.2 compares the odd-one-out accuracy of using dot product versus cosine similarity and shows that cosine similarity generally yields better alignment.

## C  CIFAR-100 TRIPLET TASK

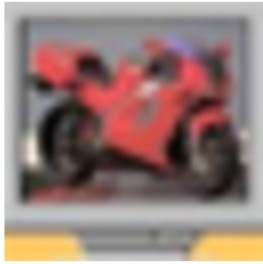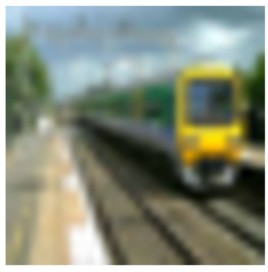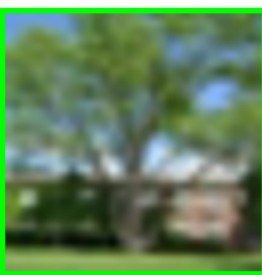

Figure C.1: An example triplet from the CIFAR-100 coarse dataset. The left two images are from one of the two CIFAR-100 "vehicle" superclasses, so the rightmost image is the odd-one-out.

In a similar vein to the THINGS triplet task, we constructed a reference triplet task from the CIFAR-100 dataset (Krizhevsky & Hinton, 2009). To show pairs of images that are similar to each other, but do not depict the same object, we leverage the 20 coarse classes of the dataset rather than the original fine-grained classes. For each triplet, we sample two images from the same and an one odd-one-out image from a different coarse class. We restrict ourselves to examples from the CIFAR-100 train set and exclude the validation set. We randomly sample a total of 50,000 triplets which is equivalent to the size of the original train set. Figure C.1 shows an example triplet for this task.

We find that ImageNet accuracy is highly correlated with odd-one-out accuracy for the CIFAR-100 coarse task (see Figure C.2; $r = 0.70$), which is in stark contrast to its correlation with accuracy on human odd-one-out judgments, which is significantly weaker (see Figure 2).

The main reason for constructing this task was to examine whether or not any findings from comparing human to neural network responses for the THINGS triplet odd-one-out task can be attributed to the nature of the triplet task. Instead of using the CIFAR-100 class labels, we specifically used the coarse super-classes that are possibly comparable to higher-level concepts that are relevant to human similarity judgments on the THINGS odd-one-out task. Hebart et al. (2020) and Muttenthaler et al. (2022) have shown that humans only use a small set of concepts

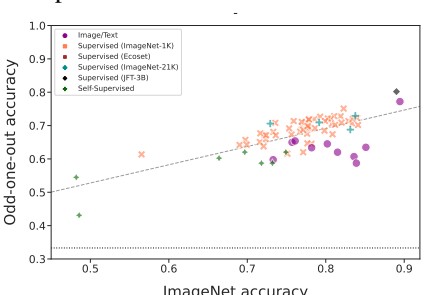

Figure C.2: Zero-shot odd-one-out accuracy on a triplet task based on CIFAR-100 coarse exhibits a strong correlation with ImageNet accuracy. Diagonal line indicates a least-squares fit.

for making similarity judgments in the triplet odd-one-out task. These concept representations are sparse representations. That is, there are only $k$ objects that are important for a concept, where $k \ll m$. Recall that $m$ denotes the number of objects in the data (e.g., 1854 for THINGS). The importance of objects is defined in Hebart et al. (2020) and Muttenthaler et al. (2022). Similarly, the coarse super-classes in CIFAR-100 are sparse. Although the CIFAR-100 triplet task may be different, we believe that additionally testing models on this task is one reasonable way to figure out whether findings (e.g., the correlation of ImageNet accuracy with triplet odd-one-out accuracy) are attributable to the nature of the triplet task itself rather than to variables related to alignment.

# D  TRANSFERABILITY OF PENULTIMATE LAYER VS. LOGITS

In Figure E.2, we show that the logits typically outperform the penultimate layer in terms of zero-shot odd-one-out accuracy. In this section, we perform a similar comparison of the performance of the penultimate layer and logits in the context of transfer learning. We find that, contrary to odd-one-out accuracy, transfer learning accuracy is consistently highest in the penultimate layer.

Following Kornblith et al. (2019b), we report the accuracy of $\ell_2$-regularized multinomial logistic regression classifiers on 12 datasets: Food-101 dataset (Bossard et al., 2014), CIFAR-10 and CIFAR-100 (Krizhevsky & Hinton, 2009), Birdsnap (Berg et al., 2014), the SUN397 scene dataset (Xiao et al., 2010), Stanford Cars (Krause et al., 2013), FGVC Aircraft (Maji et al., 2013), the PASCAL VOC 2007 classification task (Everingham et al., 2010), the Describable Textures Dataset (DTD) (Cimpoi et al., 2014), Oxford-IIIT Pets (Parkhi et al., 2012), Caltech-101 (Fei-Fei et al., 2004), and Oxford 102 Flowers (Nilsback & Zisserman, 2008). We use representations of the 16 models previously studied by Kornblith et al. (2019b) (see Table B.1), and follow the same procedure for training and evaluating the classifiers.

Results are shown in Figure D.1. For nearly all models and transfer datasets, the penultimate layer provides better representations for linear transfer than the logits layer.

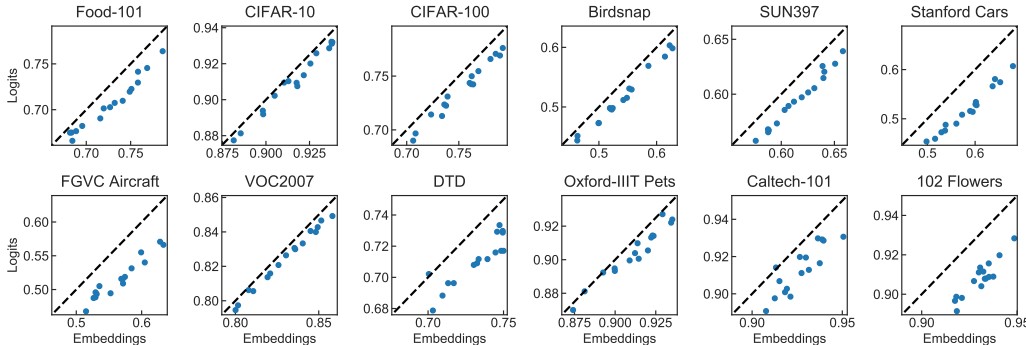

Figure D.1: Penultimate layer embeddings consistently offer higher transfer accuracy than the logits layer. Points reflect the accuracy of a multinomial logistic regression classifier trained on the penultimate layer embeddings ($x$-axis) or logits ($y$-axis) of the 16 models from Kornblith et al. (2019b), which were all trained with vanilla softmax cross-entropy. Dashed lines reflect $y = x$.

# E    LINEAR PROBING

In Figure E.1 we compare probing odd-one-out accuracy with zero-shot odd-one-out accuracy for models pretrained on ImageNet-1K or ImageNet-21K. We observe a strong positive correlation of $r = 0.963$ between probing and zero-shot odd-one-out accuracies.

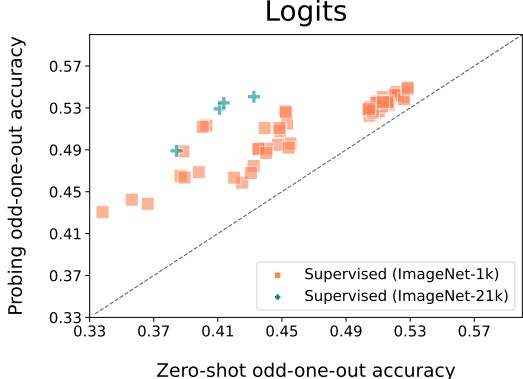

Figure E.1: Probing odd-one-out accuracy as a function of zero-shot odd-one-out accuracy for the logits layer of all ImageNet models in Table B.1.

In the top left panel of Figure E.3, we show probing odd-one-out accuracy as a function of ImageNet accuracy for all models in Table B.1. Similarly to the findings depicted in the top left panel of Figure 2, we observe a low Pearson correlation coefficient ($r = 0.213$). The remaining panels of Figure E.3 visualize probing odd-one-out accuracy as a function of ImageNet accuracy for the same subsets of models as shown in Figure 2. Again, the relationships are similar to those observed for zero-shot odd-one-out accuracy in Figure 2.

Probing accuracies show less variability than zero-shot accuracies among the networks trained with the different loss functions from Kornblith et al. (2021) (Figure 2 top center), although cosine softmax, which performed best for the zero-shot setting, is also among the best-performing models here. Moreover, probing reduced the differences between different Siamese self-supervised learning models (Figure 2 bottom left), although Siamese models still performed substantially better than the non-Siamese models. Yet, as in our analysis of zero-shot accuracy (§4.1), architecture (Figure 2 top right) or model size (Figure 2 bottom right) did not affect odd-one-out accuracy. These findings hold across every dataset we have considered in our analyses (see Figure E.4).

Whereas the logits often achieve a higher degree of alignment with human similarity judgments than the penultimate layer for zero-shot performance, alignment is nearly always highest for the penultimate layer after applying $W$ — the transformation matrix learned during linear probing — to a model's raw representation space and then performing the odd-one-out task or RSA (see Figure E.2).

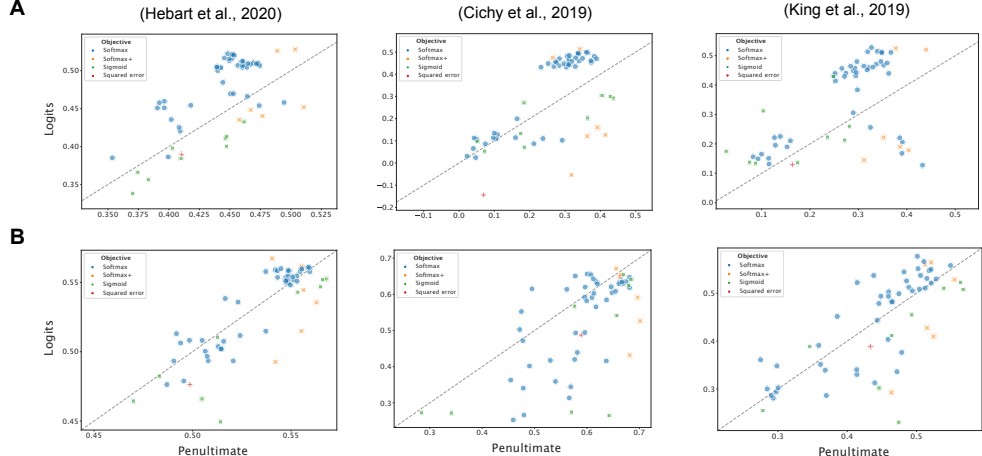

Figure E.2: Performance of the logits vs. penultimate layer for all models across all three datasets without (top row) and with (bottom row) applying the transformation matrix obtained from linear probing to a model's raw representation space. $x$-axis and $y$-axis represent odd-one-out accuracy for Hebart et al. (2020) and Spearman's $\rho$ for Cichy et al. (2019) and King et al. (2019). Networks are colored by their loss function. "softmax+" indicates softmax cross-entropy with additional regularization or normalization. Dashed lines indicate $y = x$.

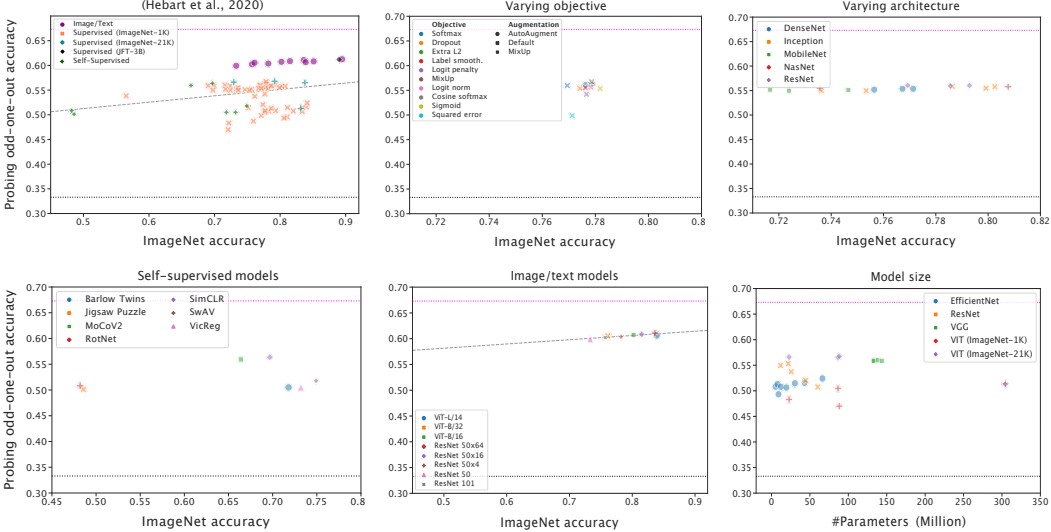

Figure E.3: Probing odd-one-out accuracy as a function of ImageNet accuracy or number of model parameters. **Top left**: Probing odd-one-out accuracies for the embedding layer of all models considered in our analysis. **Top center**: Models have the same architecture (ResNet-50) but were trained with a different objective function (Kornblith et al., 2021). **Top right**: Models were trained with the same objective function but vary in architecture (Kornblith et al., 2019b). **Bottom left**: Performance for different SSL models. **Bottom center**: Different image/text models with their ImageNet accuracies. **Bottom right** A subset of ImageNet models including their number of parameters. Dashed diagonal lines indicate a least-squares fit. Dashed horizontal lines reflect chance-level or ceiling accuracy respectively.

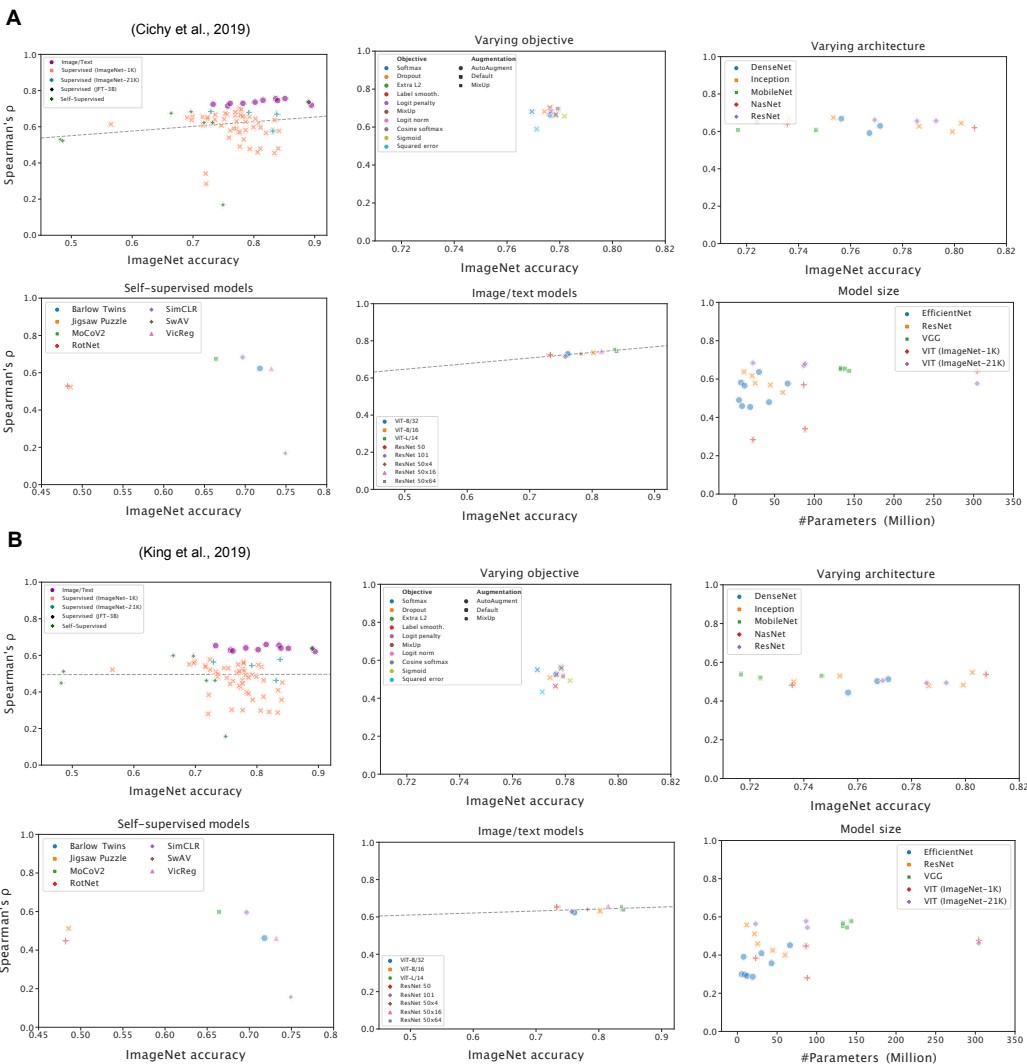

Figure E.4: Spearman rank correlation after applying $W$ to a neural net's representation space is weakly correlated with ImageNet accuracy and varies with training objective but not with model architecture or number of parameters for both human similarity judgment datasets from Cichy et al. (2019) and King et al. (2019) respectively. Diagonal lines indicate a least-squares fit.

## F LINEAR REGRESSION

In this section, we elaborate upon the results that we presented in §4.4.2 in more detail. For each of the 45 representation dimensions $j$ from VICE, we minimize the following least-squares objective

$$\arg\min_{\boldsymbol{A}_{j,:},b_j} \sum_{i=1}^{m} \underbrace{(Y_{i,j} - (\boldsymbol{A}_{j,:}\boldsymbol{x}_i + b_j))^2}_{\text{MSE}} + \alpha_j \|\boldsymbol{A}_{j,:}\|_2^2, \tag{3}$$

where $Y_{i,j}$ is the value of the $j^{\text{th}}$ VICE dimension for image $i$, $\boldsymbol{x}_i$ is the neural net representation of image $i$, and $\alpha_j > 0$ is a regularization hyperparameter. We optimize each dimension separately, selecting $\alpha_j$ via cross validation (see Appendix A.3), and assess the fit in two ways. First, we directly measure the $R^2$ for held-out images. Second, we evaluate odd-one-out accuracy of the transformed neural net representations using a similarity matrix $\boldsymbol{S}$ with $S_{ij} := (\boldsymbol{A}\boldsymbol{x}_i + \boldsymbol{b})^\top(\boldsymbol{A}\boldsymbol{x}_j + \boldsymbol{b})$, with $\boldsymbol{A}$ and $\boldsymbol{b}$ obtained from Eq. 3 (i.e., *regression odd-one-out accuracy*).

In Figure F.1, we compare odd-one-out accuracies after linear probing and regression respectively. The

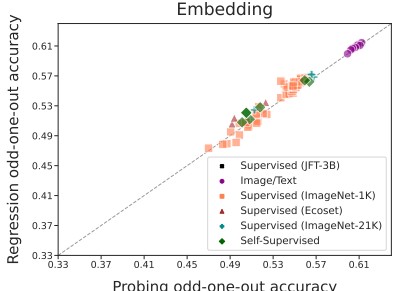

Figure F.1: Regression as a function of probing odd-one-out accuracies for all models in Table B.1.

two performance measures are highly correlated for the embedding ($r = 0.982$) and logit ($r = 0.972$; see Figure F.3) layers.[2] We provide R$^2$ values for individual concepts in Appendix F. We observe that the leading VICE dimensions, which are the most important for explaining human triplet responses, could be fitted with an R$^2$ score of $> 0.7$ for most of the models – the quality of the regression fit generally declines with the importance of a dimension (see Figure F.4).

We compared zero-shot with regression odd-one-out accuracies (as defined in §4.4.2) for the embedding layer of all models in Table B.1 and observe a strong positive relationship ($r = 0.795$; Figure F.2). In addition, we contrasted regression odd-one-out accuracy between logit and penultimate layers for ImageNet models. The results are consistent with the findings obtained from the linear probing experiments, shown in Figure 4. Moreover, we observe that probing and regression odd-one-out are highly correlated, thus applying similar transformations to a neural net's representation space. This is demonstrated in Figure F.1 for the embedding layer of all models in Table B.1 and in Figure F.3 for the logit layers of the ImageNet models. Figure F.4 shows R$^2$ scores of the linear regression fit from embedding-layer representations of all models in Table B.1 to the same subset of VICE dimensions that we leveraged for the linear probing experiments in Figure 5.

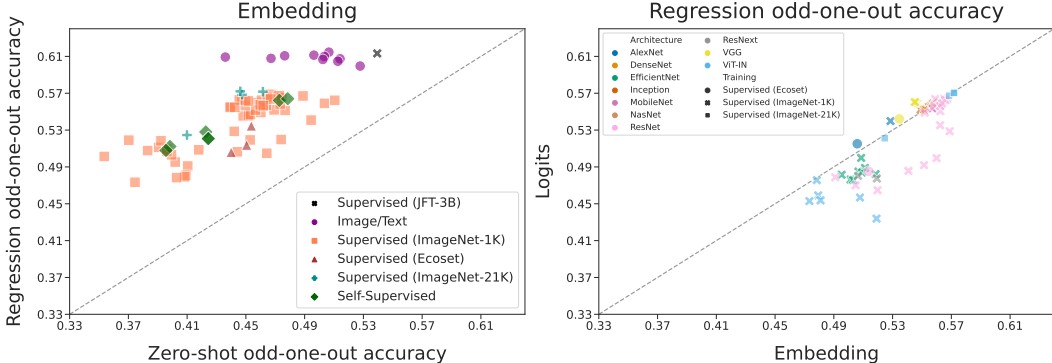

Figure F.2: **Left**: Zero-shot and regression odd-one-out accuracies for the embedding layer of all neural nets. **Right**: Regression odd-one-out accuracy for the embedding and logits layer for all supervised models trained on ImageNet-1K or ImageNet-21K.

---

[2]Note that odd-one-out accuracies are slightly higher for linear regression (Figure F.1). We hypothesize that this is because VICE is trained on all images, and thus the transformation matrix learned in linear regression has indirect access to the images it is evaluated on, whereas the linear probe has no access to these images (see Appendix A.1).

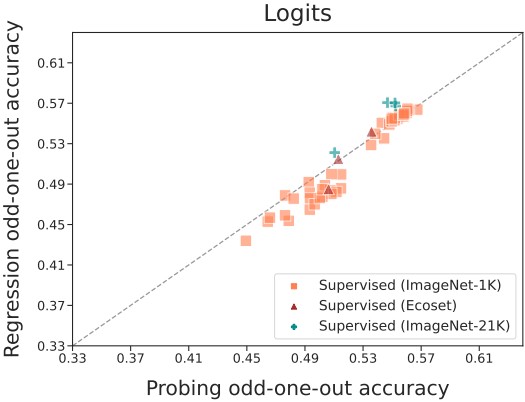

Figure F.3: Regression as a function of probing odd-one-out accuracies for the logits layers of all ImageNet models in Table B.1.

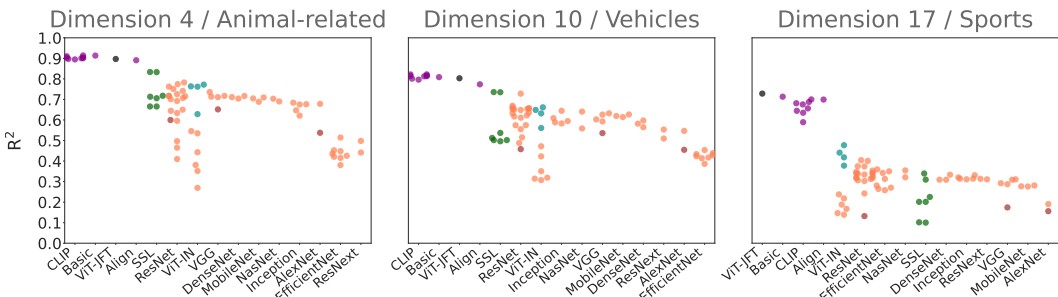

Figure F.4: $R^2$ scores for all models in Table B.1 after fitting an $\ell_2$-regularized linear regression to predict individual VICE dimensions from the embedding-layer representation of the images in THINGS. Color-coding was determined by training data/objective. Violet: Image/Text. Green: Self-supervised. Orange: Supervised (ImageNet-1K). Brown: Supervised (Ecoset). Cyan: Supervised (ImageNet-21K). **Black**: Supervised (JFT-3B).

## G  ENTROPY

Let $\mathbb{A} := \{\{i,j\}\},\{i,k\},\{j,k\}\}$ be the set of all combinations of pairs in a triplet. The entropy of a triplet can then be written as

$$H(\{i,j,k\}) = - \sum_{\{a,b\}\in\mathbb{A}} \hat{p}(\{a,b\}|\{i,j,k\}) \log \hat{p}(\{x,y\}|\{i,j,k\}),$$

where $\hat{p}(\{a,b\}|\{i,j,k\})$ is derived from the VICE model and is defined precisely in Equation 6 in Muttenthaler et al. (2022).

To understand how aleatoric uncertainty of odd-one-out predictions varies across different models, we calculated the entropy of each triplet in THINGS for every model in Table B.1 and subsequently computed the Pearson correlation coefficient of these per-triplet entropies for every pair of models.

Although models with the same architecture often correlated strongly with respect to their aleatoric uncertainty across triplets, not a single model achieves a strong positive correlation to VICE (see Figure G.1 and Figure G.2 respectively). Interestingly, the choice of the objective function appeared to play a crucial role for the entropy-alignment of a neural net with other neural nets or with VICE.

### G.1  HUMAN UNCERTAINTY DETERMINES ODD-ONE-OUT ERRORS

Since VICE provides a probabilistic model of humans' responses on the odd-one-out task, we can use the entropy of a given triplet's probability distribution to infer humans' uncertainty regarding the odd-one-out. In this section, we evaluate a model's odd-one-out classification error as a function of a triplet's entropy. For this analysis, we partitioned the original triplet dataset $\mathcal{D}$ into 11 sets $\mathcal{D}_1, \ldots, \mathcal{D}_{11}$, corresponding to 11 bins. A triplet belongs to a subset $\mathcal{D}_i$ if the entropy of the VICE

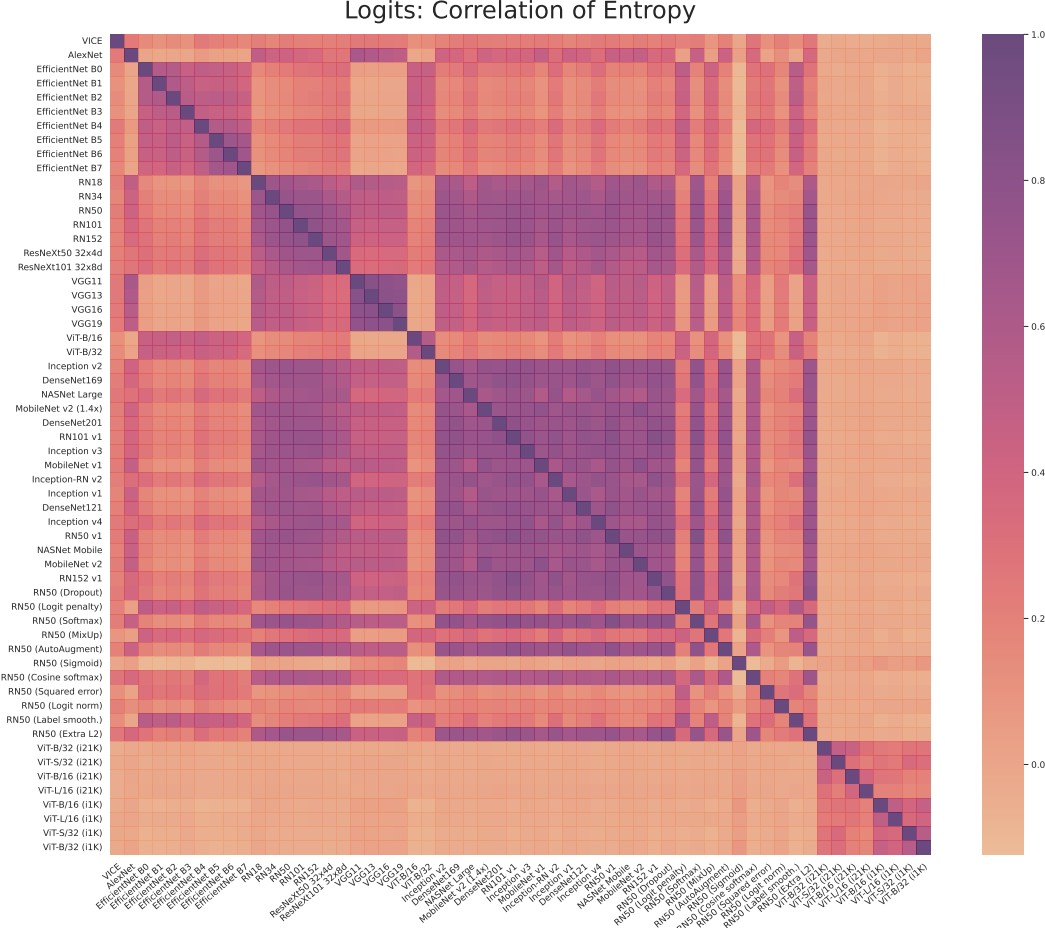

Figure G.1: The correlation coefficient of entropy of the output probabilities for each triplet in THINGS.

output distribution for that triplet falls in between the $(i-1)^{\text{th}}$ and $i^{\text{th}}$ bins' boundaries. We define a triplet's entropy as the entropy over the three possible odd-one-out responses, estimated using 50 Monte Carlo samples from VICE (see details in Appendix G). Note that the entropy for a discrete probability distribution with 3 outcomes lies in $[0, \log(3)]$.

Unsurprisingly, all models in Table B.1 yield a high zero-shot odd-one-out classification error for triplets that have high entropy, and all models' error rates increase monotonically as human triplet entropies increase. However, most models make a substantial number of errors even for triplets where entropy is low and thus humans are very certain. We find that VGGs, ResNets, EfficientNets, and ViTs trained on ImageNet-1K or ImageNet-21K and SSL models show a similarly high zero-shot odd-one-out error, between $0.1$ and $0.3$, for triplets with low entropy, whereas ALIGN and in particular CLIP-RN, CLIP-ViT, BASIC-L and ViT-G/14 JFT achieve a near-zero zero-shot odd-one-out error for the same set of triplets (see Figure G.3).

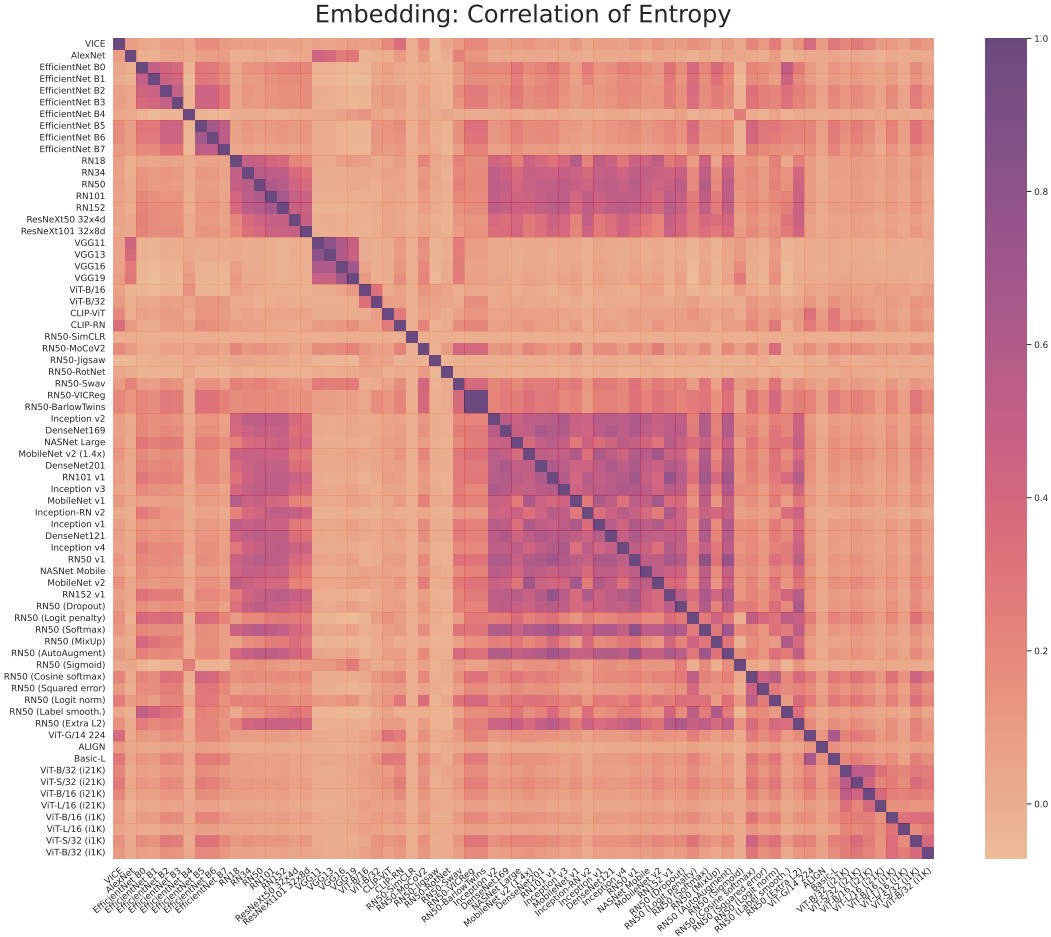

Figure G.2: The correlation coefficient of entropy of the output probabilities for each triplet in THINGS.

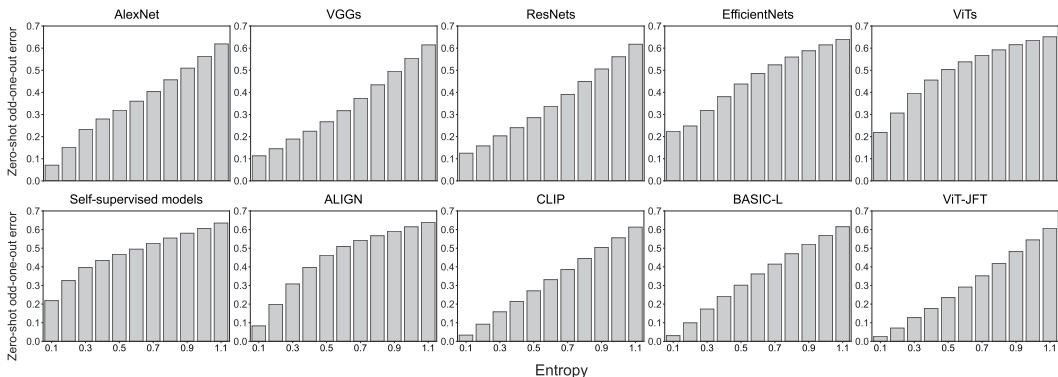

Figure G.3: Zero-shot odd-one-out prediction errors as a function of a triplet's entropy differ across model classes. **Top**: Logits layer of ImageNet supervised models. **Bottom**: Embedding layer of SSL, Image/Text models and ViT-G/14 JFT. Since models with the same architecture, trained on the same data (e.g., ImageNet-1K) with the same objective function, perform very similarly in their odd-one-out choices, we aggregated their predictions and report the average. To isolate architecture and training data from any other potentially confounding variables, we excluded models from Kornblith et al. (2021) when aggregating the ResNet predictions.

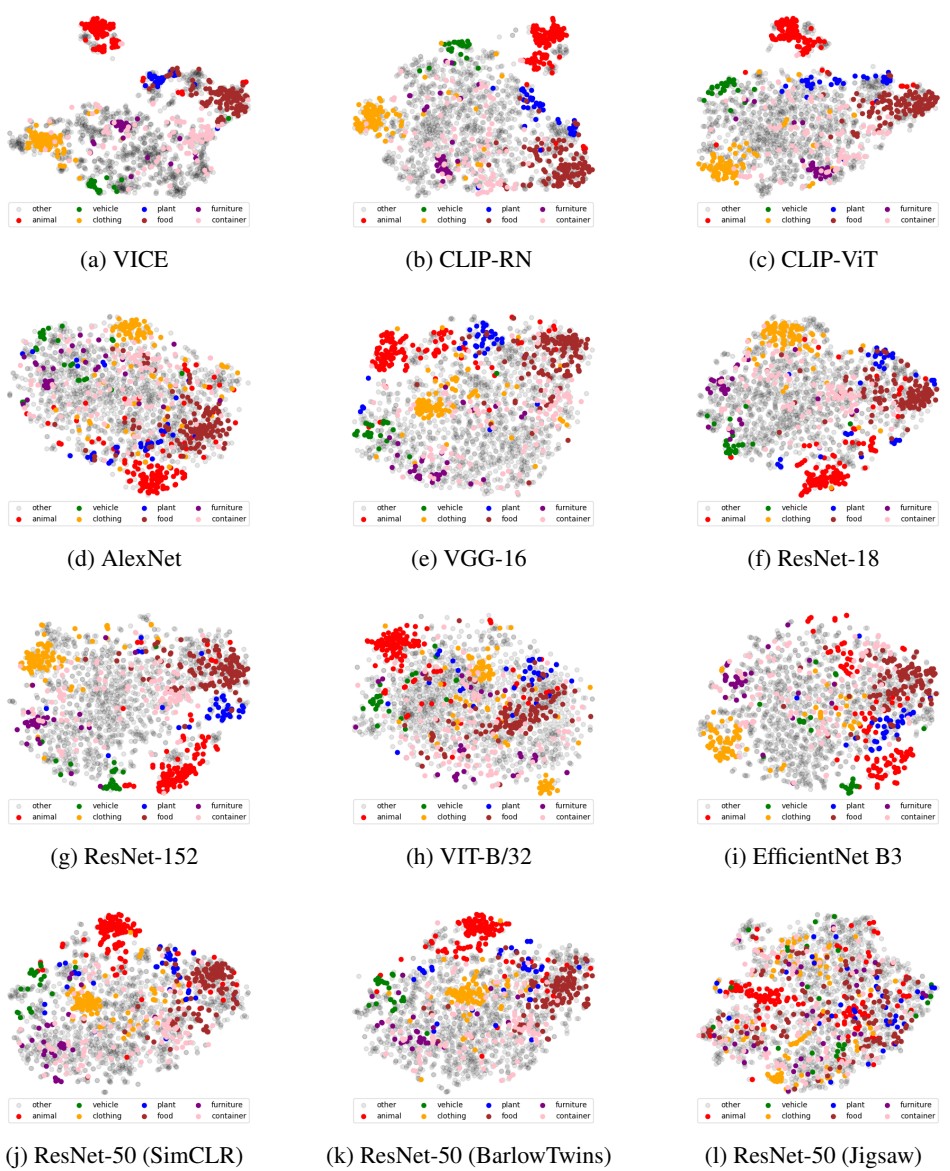

Figure H.1: t-SNE visualizations for the embedding layer of a subset of the models in Table B.1. Data points are labeled according to higher-level categories provided in the THINGS database (Hebart et al., 2019).

# H DIFFERENT CONCEPTS ARE DIFFERENTLY DISENTANGLED IN DIFFERENT REPRESENTATION SPACES

In this section, we show t-SNE (Van der Maaten & Hinton, 2008) visualizations of the embedding layers of a subset of the models in Table B.1. To demonstrate the difference in disentanglement of higher-level concepts - which are provided in the THINGS database (Hebart et al., 2019) - in different representation spaces, we have chosen a representative model for a subset of the architectures and training objectives. Figure H.1 shows that the animal concept is substantially more disentangled from other high-level categories in the representation space of image/text models such as CLIP-RN/CLIP-ViT (Radford et al., 2021) than it is for ImageNet models. The bottom row of Figure H.1 shows that higher-level concepts appear to more distributed and poorly disentangled in Jigsaw, a non-Siamese self-supervised model, compared to SimCLR and BarlowTwins, contrastive and non-contrastive self-supervised models respectively.

## I  ODD-ONE-OUT AGREEMENTS REFLECT REPRESENTATIONAL SIMILARITY

To understand whether odd-one-out choice agreements between different models reflect similarity of their representation spaces, we compared the agreement in odd-one-out choices between pairs of models with their linear Centered Kernel Alignment (CKA), a widely adopted similarity metric for neural network representations (Kornblith et al., 2019a; Raghu et al., 2021). Let $X \in \mathbb{R}^{m \times p_1}$ and $Y \in \mathbb{R}^{m \times p_2}$ be representations of the same $m$ examples obtained from two neural networks with $p_1$ and $p_2$ neurons respectively. Assuming that the column (neuron) means have been subtracted from each representation (i.e., centered representations), linear CKA is defined as

$$\mathrm{CKA}_{\mathrm{linear}}(\boldsymbol{X}, \boldsymbol{Y}) = \frac{\mathrm{vec}(\boldsymbol{X}\boldsymbol{X}^\top) \cdot \mathrm{vec}(\boldsymbol{Y}\boldsymbol{Y}^\top)}{\|\boldsymbol{X}\boldsymbol{X}^\top\|_{\mathrm{F}} \|\boldsymbol{Y}\boldsymbol{Y}^\top\|_{\mathrm{F}}}.$$

Intuitively, the representational similarity (Gram) matrices $\boldsymbol{X}\boldsymbol{X}^\top$ and $\boldsymbol{Y}\boldsymbol{Y}^\top$ measure the similarity between representations of different examples according to the representations contained in $\boldsymbol{X}$ and $\boldsymbol{Y}$. Linear CKA measures the cosine similarity between these representational similarity matrices after flattening them to vectors.

In Figure I.1 we show heatmaps for both zero-shot odd-one-out agreements and CKA for the same pairs of models. The regression plot shows zero-shot odd-one-out choice agreement between all pairs of models in Table B.1 as a function of their CKA. We observe a high correlation between odd-one-out choice agreements and CKA for almost every model pair. That is, odd-one-out choice agreements appear to reflect similarities of neural network representations.

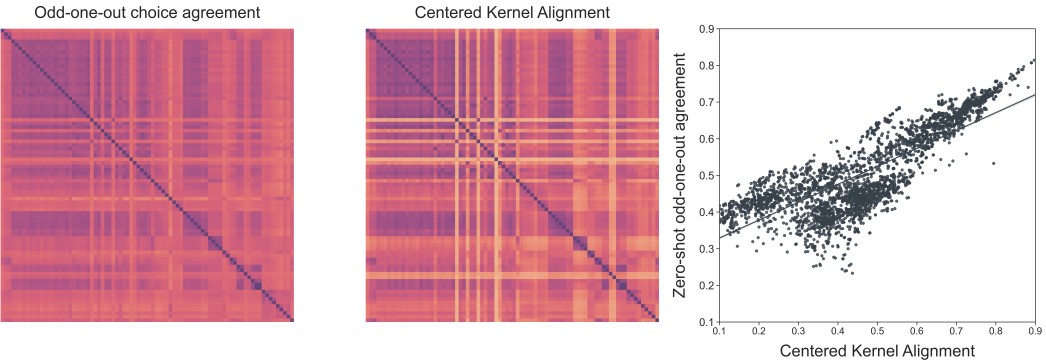

Figure I.1: **Left**: Heatmaps for zero-shot odd-one-out choice agreements and CKA for the same pairs of models. For better readability, we omitted labeling the names of the individual models. **Right**: Zero-shot odd-one-out choice agreements between all pairs of models as a function of CKA plus a linear regression fit.

## J  HUMAN ALIGNMENT IS CONCEPT SPECIFIC

In Figure J.1 and Figure J.2 we compare zero-shot odd-one-out accuracy and $\mathrm{R}^2$ scores respectively between the best-aligned ImageNet model and the best-aligned overall model. Although ViT-G/14 JFT achieves better alignment with human similarity judgments for most VICE dimensions, the best ImageNet model outperformed ViT-G/14 JFT for a small subset of the concepts, e.g., `tools`, `technology`, `paper`, `liquids`. That is, there appear to be some human concepts which ImageNet models can represent fairly well without an additionally learned linear transformation matrix, even better than a model trained on a larger, more diverse dataset. However, the $\mathrm{R}^2$ scores show a different pattern. In linear regression, representations of ViT-G/14 JFT could clearly be fitted better to the VICE dimensions for every concept compared to the best ImageNet model.

In Figure J.3 we show per-concept zero-shot and probing odd-one-out accuracies for all models in Table B.1 for all 45 VICE dimensions. Whereas zero-shot odd-one-out performances did not show a consistent pattern across the individual dimensions, probing odd-one-out performances clearly demonstrated that image/text models and ViT-G/14 JFT are better aligned with human similarity judgments for almost every concept. However, there are some concepts (e.g., outdoors-related objects/dimension 8; powdery/dimension 22) for which these models are worse aligned than ImageNet models. The difference in human alignment between image/text models plus ViT-G/14 JFT and ImageNet models is largest for royal and sports-related objects - i.e, dimension 7 and 17 respectively

-, where image/text models and ViT-G/14 JFT outperformed ImageNet models by a large margin. The bottom panel of the figure shows $R^2$ scores of the regression fit for each VICE dimension using the embedding layer representations of all models in Table B.1. We observe that more important concepts are generally easier to recover. Recall that dimensions are numbered according to their importance (Muttenthaler et al., 2022).

In addition, Figure J.4 - Figure J.7 show both zero-shot and probing odd-one-out accuracies for all models in Table B.1 for a larger set of human concepts as we do in the main text, including the 6 most important THINGS images/categories for the dimensions themselves.

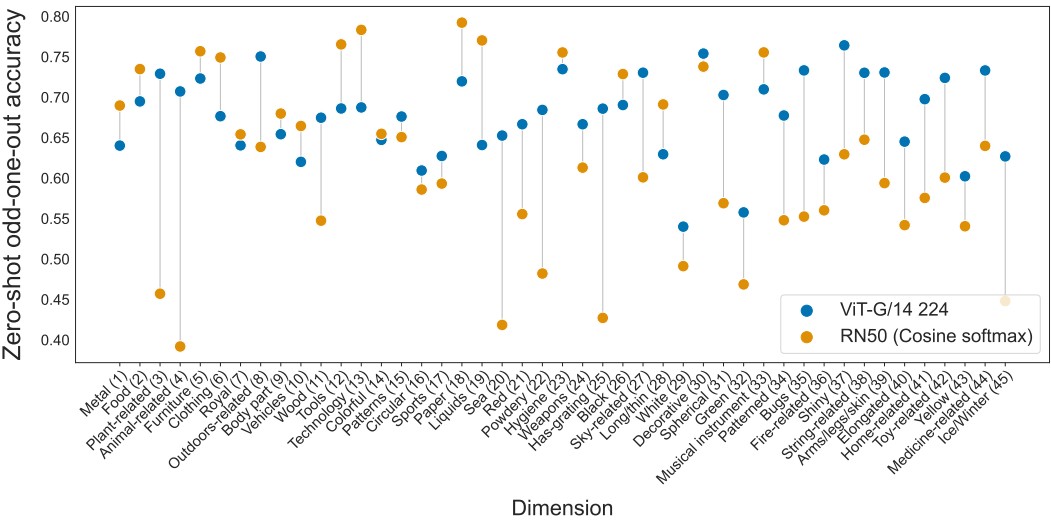

Figure J.1: Comparison of zero-shot odd-one-out accuracy between the best ImageNet and the best overall model for all 45 VICE dimensions.

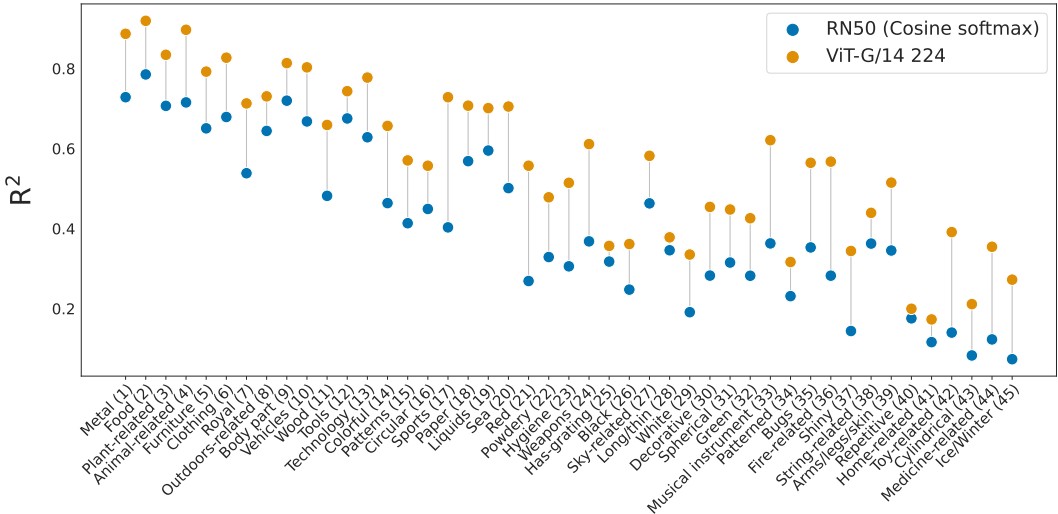

Figure J.2: Comparison of the regression $R^2$-scores between the best ImageNet and the best overall model for all 45 VICE dimensions.

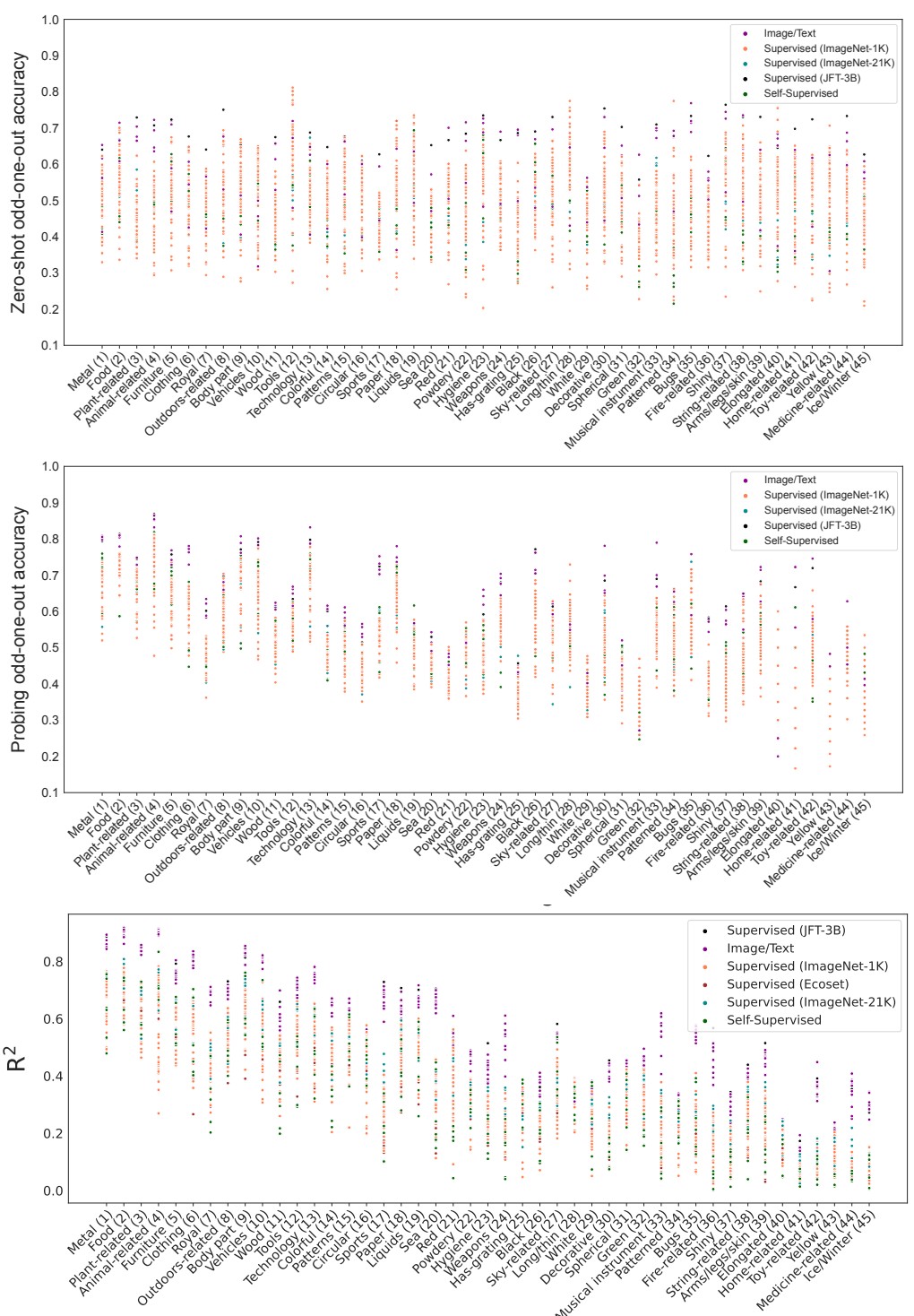

Figure J.3: **Top**: Zero-shot odd-one-out accuracies of all models in Table B.1 for all 45 VICE dimensions. **Middle**: Probing odd-one-out accuracies of all models in Table B.1 for all 45 VICE dimensions. **Bottom**: $R^2$ scores for all models in Table B.1 after fitting an $\ell_2$-regularized linear regression to predict individual VICE dimensions/human concepts.

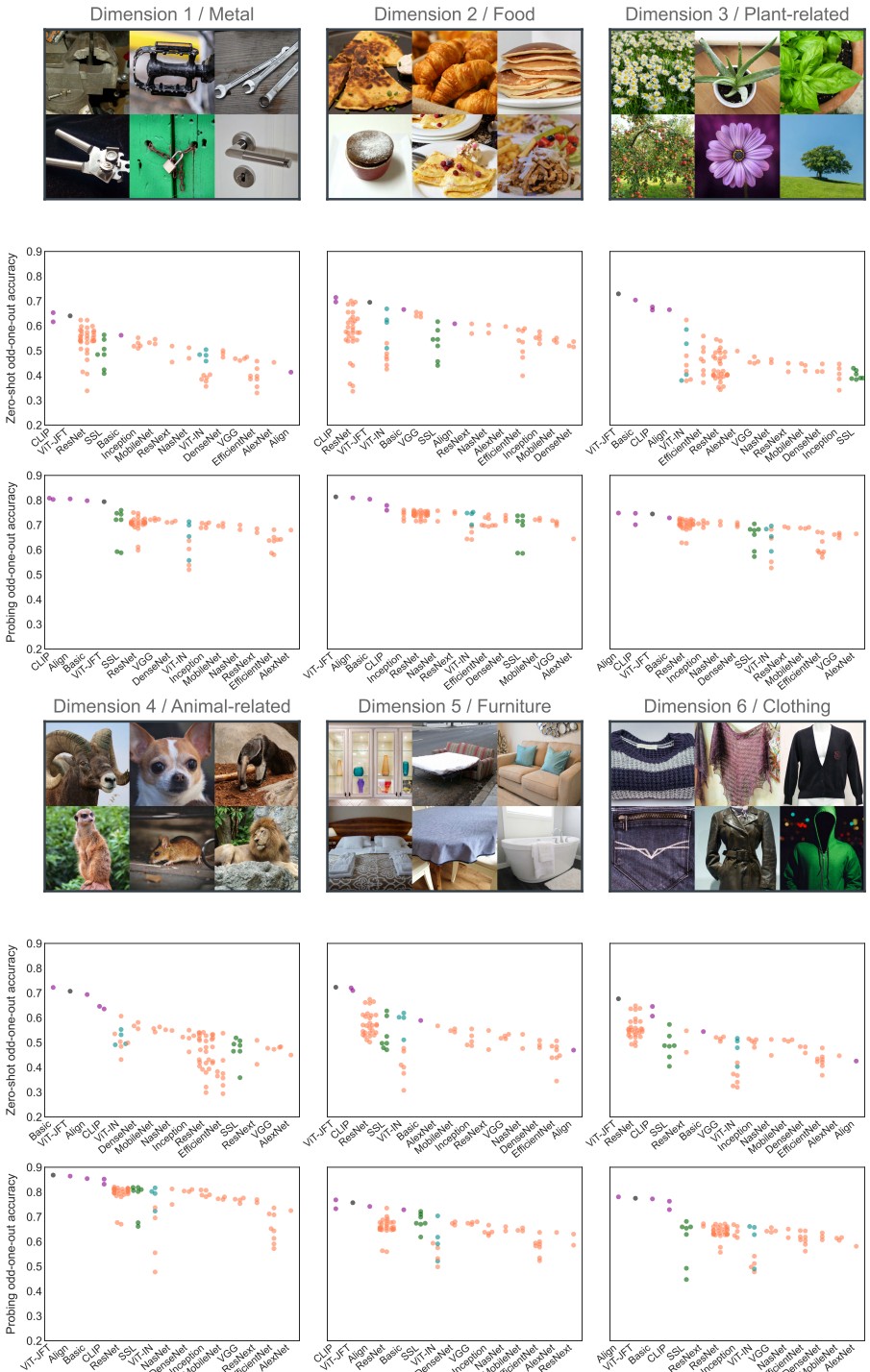

Figure J.4: Zero-shot and probing accuracy for triplets discriminated by VICE dimensions 1-6, following the approach described in §4.4. Color-coding is determined by training data/objective. Violet: Image/Text. Green: Self-supervised. Orange: Supervised (ImageNet-1K). Cyan: Supervised (ImageNet-21K). **Black**: Supervised (JFT-3B).

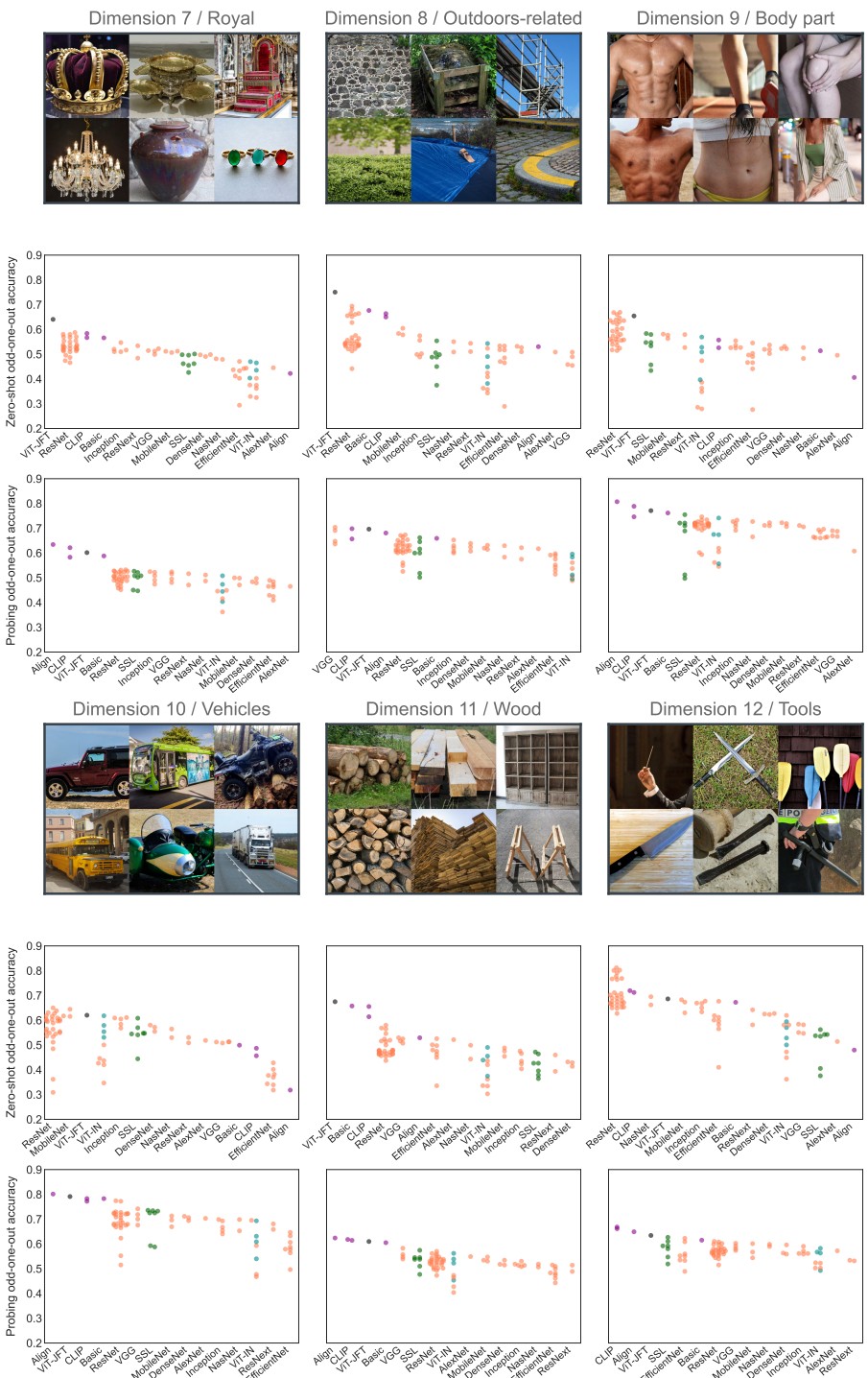

Figure J.5: Zero-shot and probing accuracy for triplets discriminated by VICE dimensions 7-12, following the approach described in §4.4. Color-coding is determined by training data/objective. Violet: Image/Text. Green: Self-supervised. Orange: Supervised (ImageNet-1K). Cyan: Supervised (ImageNet-21K). **Black**: Supervised (JFT-3B).

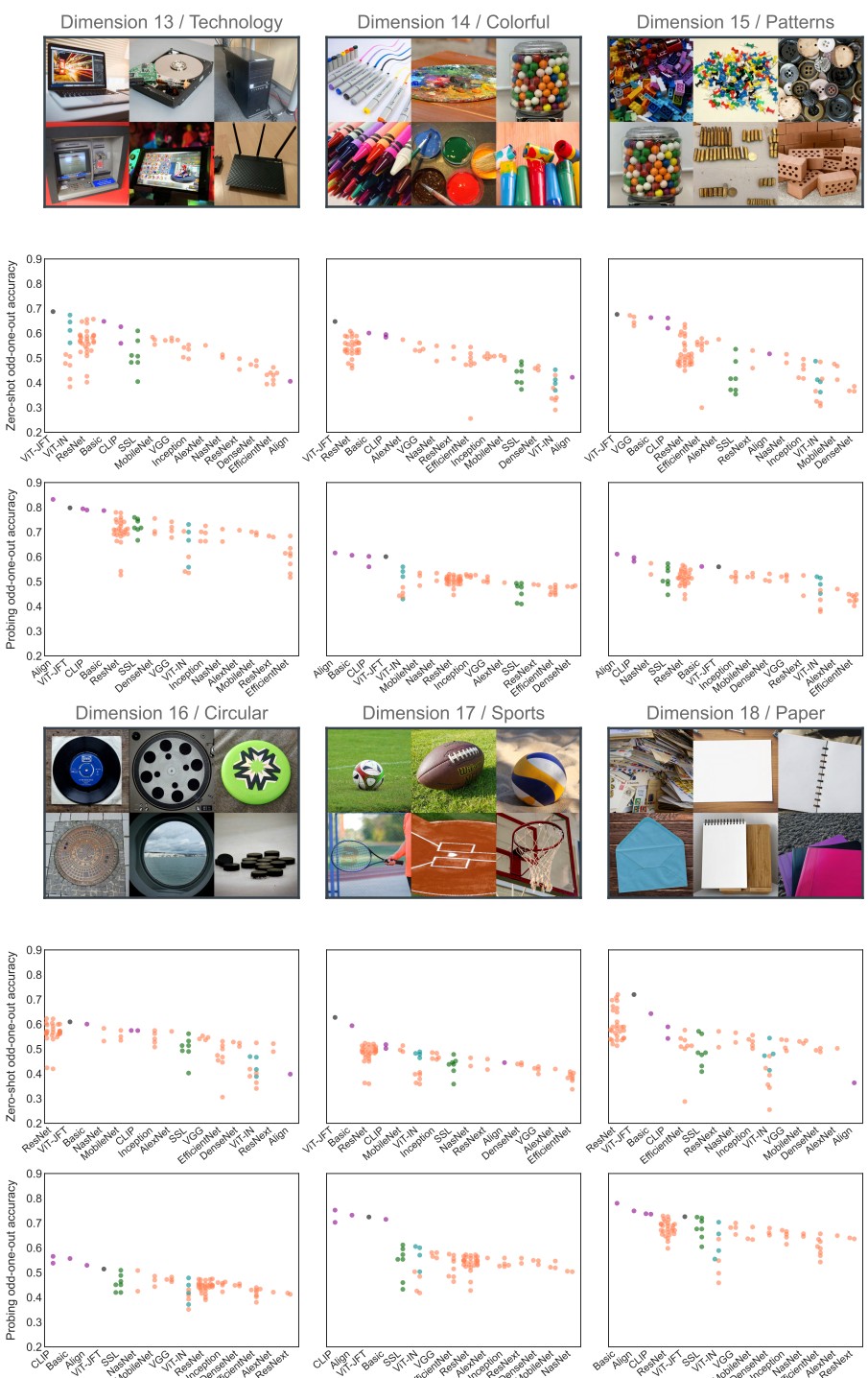

Figure J.6: Zero-shot and probing accuracy for triplets discriminated by VICE dimensions 13-18, following the approach described in §4.4.Color-coding is determined by training data/objective. Violet: Image/Text. Green: Self-supervised. Orange: Supervised (ImageNet-1K). Cyan: Supervised (ImageNet-21K). **Black**: Supervised (JFT-3B).

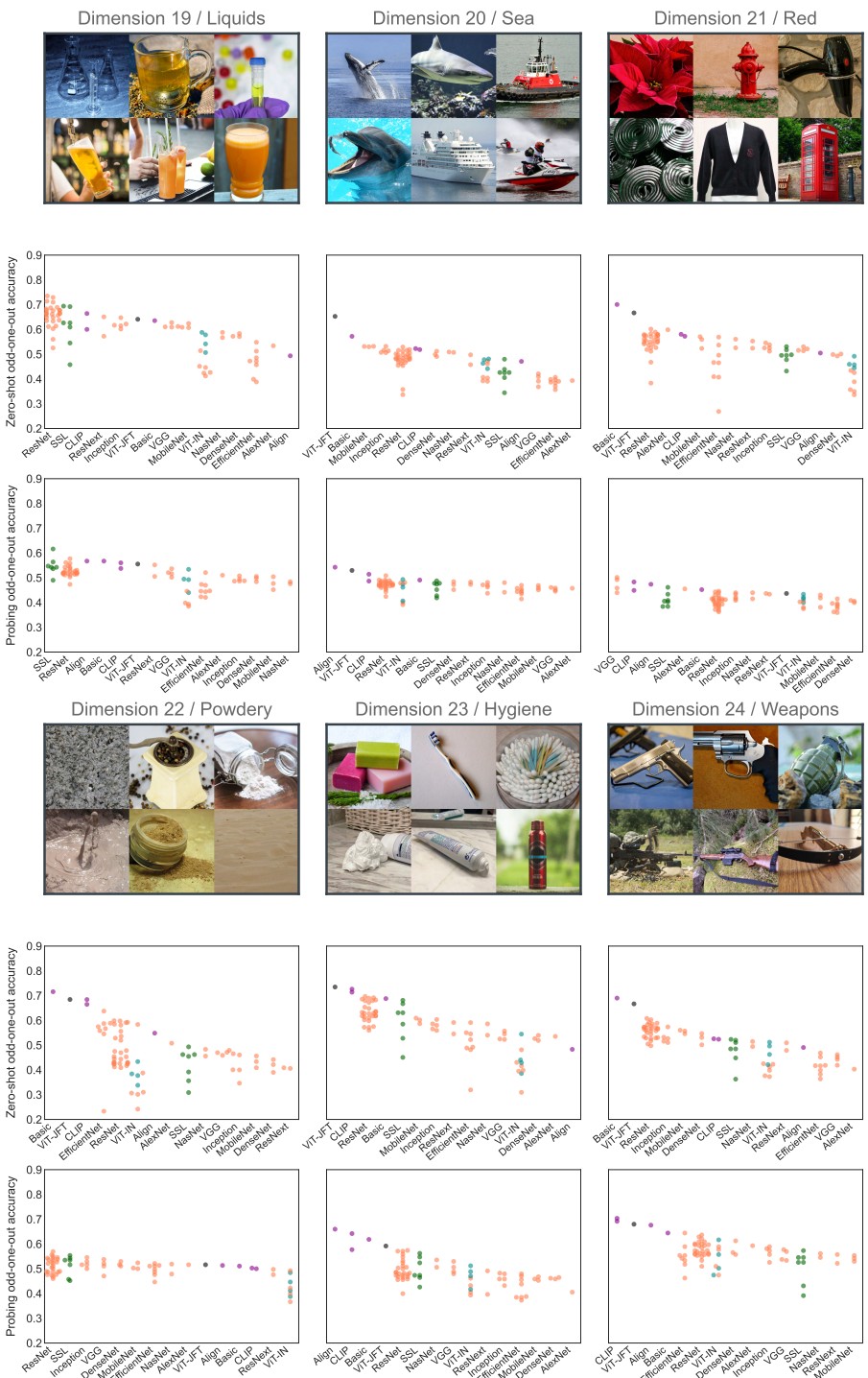

Figure J.7: Zero-shot and probing accuracy for triplets discriminated by VICE dimensions 19-24, following the approach described in §4.4.Color-coding is determined by training data/objective. Violet: Image/Text. Green: Self-supervised. Orange: Supervised (ImageNet-1K). Cyan: Supervised (ImageNet-21K). **Black**: Supervised (JFT-3B).

