# OpenReview forum: "Human alignment of neural network representations"
_ICLR.cc/2023/Conference — ICLR 2023 poster_

### Official Review · Reviewer_u84n · 2022-10-24

**Confidence:** 4
**Correctness:** 4
**Technical Novelty And Significance:** 2
**Empirical Novelty And Significance:** 3
**Recommendation:** 8

**Clarity, Quality, Novelty And Reproducibility:**

Very clear writing.

The work involves running an existing dataset (THINGS) through pre-trained DNNs. It seems to be the first time this has been done but the question/answers here do not feel very novel.

The work looks to be highly reproducible.

**Strength And Weaknesses:**

Strengths
- Awesome introduction.
- I am all for evaluations that assess how well today's models are explaining human perception, so I appreciate that contribution.


Weaknesses
- There's no finding here that moves the field of cog/vis sci or AI. The authors look at odd-one-out accuracy on the things dataset. They test a bunch of models. Some do better than others.
- The potentially most interesting finding is that "scaling ImageNet
models does not lead to better alignment of their representations with human similarity judgments." However, the authors also found that pretraining models on bigger-than-imagenet datasets (like JIT-300) does improve alignment. Of course those large-data models are also bigger, so the deleterious effect of scale that they found can be easily counteracted by pursuing the scaling laws that have taken hold of the field today — there's no novel prescription that falls out of this work.
- The gap that the authors note between the optimal score of 67% and models' scores of 57-58% is numerically small but potentially important. However, unlike similar work like Geirhos et al., 2021 "Partial success in closing the gap..." which focused on OOD generalization for humans and models, it's a stretch to find a similar utility for the things dataset triplet scores.

- The analysis on what model specs affect odd-one-out accuracy could be refined. "Varying objective" is a combination of varying losses, augmentations, and regularizations. "Varying architecture" similarly has a lot of stuff going on at once. "Model size" is not just model size, but rather confounded with the many different bells-and-whistles in ResNet vs. VIT vs. VGG. I think this analysis should be redone. Choose one model architecture, like ResNet, and in separate plots, look at how different losses affect performance vs. different regularizations vs. different augmentations vs. varying depth vs. varying width vs. varying self supervision objective. Right now there's too many hidden differences between the models that go into each of the subplots in Fig. 3 to interpret those results.
- The authors describe odd-one-out accuracy as being higher in the final vs. penultimate layer of the network. It would be helpful to understand why this finding is significant. Do we learn anything about human vision based on which layer the accuracy is the highest?

Minor
- The results are written to sound like extended figure captions ("The plot in the bottom left corner of Figure 3 compares..."). It would help readability if the results and findings were woven into the overall narrative, and figures were referenced parenthetically.
- I think it would help interpretability if you plotted scores normalized by the optimal performance. In other words, models achieve X% of the optimal score
- "Prior work has investigated this question indirectly, by measuring models’ error consistency with humans (Geirhos et al., 2018; Rajalingham et al., 2018; Geirhos et al., 2021) and the ability of their representations to predict neural activity in primate brains (Yamins et al., 2014; Guc¸l ¨ u &¨ van Gerven, 2015; Schrimpf et al., 2020), with mixed results." Can you add a sentence clarifying what you mean by mixed results.




**Summary Of The Paper:**

The authors investigated the alignment between DNNs' representations of objects in the THINGS dataset and humans' judgments of those same images. They found that the newest and largest-scale models trained on larger-than-imagenet datasets like the JIT-300 or trained with caption embeddings (e.g., CLIP) performed the best. They complement this finding with several surveys to identify the architectural/training choices that drive performance on THINGS triplet predictions, and analyze the concepts captured by DNN representations.


Updated:

The additional experiments done over the rebuttal period strengthen this paper and make it a good and rigorous contribution for AI and the brain sciences.

**Summary Of The Review:**

I think the paper is either a preliminary report pointing towards bigger questions about understanding human vision and how DNNs can be improved as models of human vision or, in its current form, a solid workshop paper. There is nothing here that changes my thinking about human or machine vision and so I do not feel strongly about supporting its acceptance into ICLR.

---

> ### Author Response · Authors · 2022-11-11
> **Author response (1/3)**
>
> **There's no finding here that moves the field of cog/vis sci or AI. The authors look at odd-one-out accuracy on the things dataset. They test a bunch of models. Some do better than others.**
>
>
> We believe that it is important that the field routinely engages in the kind of evaluation that we have performed because, without such evaluation, it is not clear how developments in ML relate to topics of interest in cognitive science. We further believe that our evaluation has revealed some nontrivial findings that were not previously known:
>
> - Strong performance on computer vision tasks does not necessarily mean that a model has learned a concept space that is aligned with a human’s concept space. (We don’t claim that this is not a useful concept space but it is just not one that is aligned with human similarity judgments.)
> - Scaling model size in isolation does not lead to a better alignment of models’ representations with human behavior.
> - Differences in the objective function and training data impact human alignment, whereas differences in architecture and model size/data don’t.
> - ImageNet models make a substantial number of mistakes for triplets where humans consistently choose the same odd-one-out.
>
> These findings may help cognitive scientists to isolate the variables that most influence the alignment between human and neural network similarity judgments (e.g., the specific objective function is more important than the architecture). Given our results, it does not seem straightforward to improve human alignment by modifying the architecture, whereas increasing the diversity of the training data or changing the training objective appears to make a difference. This could help guide machine learning researchers if they are interested in developing models whose representation spaces are aligned with human similarity judgments.
>
> **The potentially most interesting finding is that "scaling ImageNet models does not lead to better alignment of their representations with human similarity judgments." However, the authors also found that pretraining models on bigger-than-imagenet datasets (like JIT-300) does improve alignment. Of course those large-data models are also bigger, so the deleterious effect of scale that they found can be easily counteracted by pursuing the scaling laws that have taken hold of the field today — there's no novel prescription that falls out of this work.**
>
> Scaling laws from previous work relate accuracy to both model size and dataset size, finding non-negligible impacts of each (e.g. [1-3]). Here, we show that odd-one-out accuracy is essentially flat as a function of model size. For example, a ResNet-152 has substantially more parameters than a ResNet-18 but our results indicate that these models don’t differ in their odd-one-out accuracy (see Table B.1). Moreover, CLIP ResNet-50 and ViT-B/32, which are relatively small models trained on a large, diverse image/text dataset, achieve substantially higher odd-one-out accuracy than architecturally identical ImageNet-trained ResNet-50 and ViT-B/32, but perform similarly to ALIGN and Basic-L, which are much larger models trained on image/text data, and to ViT-G/14, which is among the largest Vision Transformers ever trained. Thus, our results demonstrate that scaling laws for alignment are quite different from the scaling laws commonly reported for accuracy on standard computer vision tasks (e.g. those of [2]).
>
> **The gap that the authors note between the optimal score of 67% and models' scores of 57-58% is numerically small but potentially important. However, unlike similar work like Geirhos et al., 2021 "Partial success in closing the gap..." which focused on OOD generalization for humans and models, it's a stretch to find a similar utility for the things dataset triplet scores.**
>
> We believe that it’s worth asking whether, as we train ever-larger models on ever-larger datasets, we are converging on human-like representations. Answering this question clearly does not lead to improved performance on standard ML benchmark tasks in its own right. However, establishing that the answer to this question is no provides an impetus for future investigation into the properties of human representations that neural network representations currently lack, which may eventually lead to better task performance. Thus, although odd-one-out accuracy is not a practical benchmark, it is a concrete measurement that is not entirely disconnected from quantities of practical interest.

---

> > ### Author Response · Authors · 2022-11-11
> > **Author response (2/3)**
> >
> > **The analysis on what model specs affect odd-one-out accuracy could be refined. "Varying objective" is a combination of varying losses, augmentations, and regularizations. "Varying architecture" similarly has a lot of stuff going on at once. "Model size" is not just model size, but rather confounded with the many different bells-and-whistles in ResNet vs. VIT vs. VGG. I think this analysis should be redone. Choose one model architecture, like ResNet, and in separate plots, look at how different losses affect performance vs. different regularizations vs. different augmentations vs. varying depth vs. varying width vs. varying self supervision objective. Right now there's too many hidden differences between the models that go into each of the subplots in Fig. 3 to interpret those results.**
> >
> > We agree that in the “varying objective” plot objective function, augmentations, and regularization differ between the points. Since the regularizers are applied either to the penultimate or output layer, it is not clear whether there is a meaningful line to be drawn between changing the objective and adding a regularizer. However, it is a fair point that augmentation is likely to have a different impact. To improve interpretability, we have replaced the original “varying objective” plot in Figure 3 with a plot that uses color to distinguish objective function(s)/regularizers and style to distinguish between augmentation strategies. Note that the overall findings did not change.
> >
> > We are not sure what else is going on in the “varying architecture” plot aside from varying the architecture. All of the models in Kornblith et al. (2019) [4] were trained with the same objective function (softmax cross-entropy error) and the same hyperparameters (learning rate, optimizer, regularization, etc.). The only variable that differs between the points in the “varying architecture” plot is the architecture. Given that varying the entire architecture, including width and depth, has no meaningful impact on odd-one-out accuracy, we do not see how varying only width or depth could lead to different conclusions.
> >
> >
> > The “model size” plot is intended to permit comparison between members of the same model family at different sizes (i.e., points that are the same color), where the models are controlled in terms of training settings, data, etc., and not to provide meaningful comparisons across families. In our original submission, we stated that “Although results here are comparable for different model sizes, they may not be comparable across architectures, because different architectures have been trained with different hyperparameters and objectives, which both affect alignment.” We have now added a similar sentence to the caption of Figure 3: “Note that, because models belonging to different families come from different sources and were trained with different objectives, hyperparameters, etc., models are only directly comparable within a family.”
> >
> > **The authors describe odd-one-out accuracy as being higher in the final vs. penultimate layer of the network. It would be helpful to understand why this finding is significant. Do we learn anything about human vision based on which layer the accuracy is the highest?**
> >
> > Since we wanted to obtain the highest degree of alignment possible, we have done this analysis mainly to figure out which layer one is supposed to use when measuring alignment. Lots of recent studies that compared ImageNet model representations with human behavior (or other cognitive data) used the penultimate layer [5-9]. Therefore, we were interested in examining whether there is another layer in ImageNet models that possibly shows better alignment than the penultimate layer for the triplet odd-one-out task. The only conclusion that we can draw from this finding is that for ImageNet models it does not seem to matter whether to use the penultimate or logits layer. Although zero-shot performance of the logits layer generally is better, information can be recovered from the penultimate layer via a linear transform. It has never been our goal to make any claims about human vision based on this finding.
> > We believe that performing this investigation was necessary to adhere to scientific rigor, and we still find the fact that the logits layer performs better in terms of zero-shot odd-one-out accuracy somewhat puzzling in light of previous work. Nonetheless, we agree with the reviewer that this finding is not a core finding of our study and we have reduced its emphasis.
> >
> > **The results are written to sound like extended figure captions ("The plot in the bottom left corner of Figure 3 compares..."). It would help readability if the results and findings were woven into the overall narrative, and figures were referenced parenthetically.**
> >
> > Thanks for pointing this out. We agree and have changed this accordingly.

---

> > > ### Author Response · Authors · 2022-11-11
> > > **Author response (3/3)**
> > >
> > > **I think it would help interpretability if you plotted scores normalized by the optimal performance. In other words, models achieve X% of the optimal score**
> > >
> > > We considered plotting normalized accuracy but decided that plotting the raw values was the more defensible option. Nonetheless, we acknowledge the reviewer’s concern that it is difficult to interpret accuracy as we plot it. We have thus updated our main plots in Figures 2 and 3 to include horizontal lines indicating the maximum achievable accuracy as well as chance accuracy.
> > >
> > >
> > > **“Prior work has investigated this question indirectly, by measuring models’ error consistency with humans (Geirhos et al., 2018; Rajalingham et al., 2018; Geirhos et al., 2021) and the ability of their representations to predict neural activity in primate brains (Yamins et al., 2014; Guc¸l ¨ u &¨ van Gerven, 2015; Schrimpf et al., 2020), with mixed results." Can you add a sentence clarifying what you mean by mixed results.**
> > >
> > > We have added the sentence “Networks trained on more data make somewhat more human-like errors (Geirhos et al., 2021), but do not necessarily obtain a better fit to brain data (Schrimpf et al., 2020).”
> > >
> > > **Final Note**
> > >
> > > We thank the reviewer for their suggestions to improve the manuscript and hope that they will raise their score in light of our responses.
> > >
> > > **References**
> > >
> > > [1] Kaplan, J., McCandlish, S., Henighan, T., Brown, T. B., Chess, B., Child, R., ... & Amodei, D. (2020). Scaling laws for neural language models. arXiv preprint arXiv:2001.08361.
> > >
> > > [2] Zhai, X., Kolesnikov, A., Houlsby, N., & Beyer, L. (2022). Scaling vision transformers. In Proceedings of the IEEE/CVF Conference on Computer Vision and Pattern Recognition (pp. 12104-12113).
> > >
> > > [3] Hoffmann, J., Borgeaud, S., Mensch, A., Buchatskaya, E., Cai, T., Rutherford, E., ... & Sifre, L. (2022). Training Compute-Optimal Large Language Models. arXiv preprint arXiv:2203.15556.
> > >
> > > [4] Kornblith, S., Shlens, J., and Le Q. V. (2019). "Do better imagenet models transfer better?." Proceedings of the IEEE/CVF conference on computer vision and pattern recognition.
> > >
> > > [5] Peterson, J. C., Abbott, J. T., & Griffiths, T. L. (2016). Adapting deep network features to capture psychological representations. arXiv preprint arXiv:1608.02164.
> > >
> > > [6] Peterson, J. C., Abbott, J. T., & Griffiths, T. L. (2018). Evaluating (and improving) the correspondence between deep neural networks and human representations. Cognitive Science, 42(8), 2648-2669.
> > >
> > > [7] Muttenthaler, L., & Hebart, M. N. (2021). THINGSvision: a Python toolbox for streamlining the extraction of activations from deep neural networks. Frontiers in Neuroinformatics, 45.
> > >
> > > [8] Storrs, K. R., Kietzmann, T. C., Walther, A., Mehrer, J., & Kriegeskorte, N. (2021). Diverse deep neural networks all predict human inferior temporal cortex well, after training and fitting. Journal of Cognitive Neuroscience, 33(10), 2044-2064.
> > >
> > > [9] Kaniuth, P., & Hebart, M. N. (2022). Feature-reweighted representational similarity analysis: A method for improving the fit between computational models, brains, and behavior. NeuroImage, 119294.

---

> > > > ### Comment · Reviewer_u84n · 2022-11-17
> > > > **Response**
> > > >
> > > > > horizontal lines indicating the maximum achievable accuracy as well as chance accuracy
> > > >
> > > > That helps! Maybe use a different color/plotting style to differentiate these more though? Just a minor comment.
> > > >
> > > > > We have added the sentence
> > > >
> > > > Great.
> > > >
> > > >
> > > >
> > > > OVERALL:
> > > >
> > > > I'm going to raise my score. I think this work is missing that additional computational analysis (e.g., extending beyond THINGS) that I mentioned in my first response, which would make it a slam dunk. But it is indeed intriguing and well-done work that may signify a new direction in the field of comp cog neuro towards embracing and investigating the scaling laws which are driving performance in AI.

---

> > > > > ### Author Response · Authors · 2022-11-25
> > > > > **Author response**
> > > > >
> > > > > > That helps! Maybe use a different color/plotting style to differentiate these more though? Just a minor comment.
> > > > >
> > > > > Thanks for the comment. We’ve used a different color (magenta) for the upper horizontal line and updated all plots accordingly (see revised version of our manuscript).
> > > > >
> > > > > > I'm going to raise my score. I think this work is missing that additional computational analysis (e.g., extending beyond THINGS) that I mentioned in my first response, which would make it a slam dunk. But it is indeed intriguing and well-done work that may signify a new direction in the field of comp cog neuro towards embracing and investigating the scaling laws which are driving performance in AI.
> > > > >
> > > > > Thank you for your overall positive feedback. Since we like to shoot for slam dunks, we’ve added the same set of analyses that we have performed for the THINGS triplet odd-one-out task for an additional human similarity judgment dataset [1]. We find very similar results for the other similarity judgments data by performing RSA rather than measuring triplet odd-one-out accuracy. What’s more, we find that the transformation matrix obtained from the linear probe — which was trained on the THINGS triplet odd-one-out task — improves alignment for the other data. We believe that these additional results bolster our findings and make the conclusions that we have drawn previously more generalizable.
> > > > >
> > > > > **References**
> > > > >
> > > > > [1] Cichy, R. M., Kriegeskorte, N., Jozwik, K. M., van den Bosch, J. J., & Charest, I. (2019). The spatiotemporal neural dynamics underlying perceived similarity for real-world objects. Neuroimage, 194, 12-24.\
> > > > > [2] Kriegeskorte, N., & Mur, M. (2012). Inverse MDS: Inferring dissimilarity structure from multiple item arrangements. Frontiers in Psychology, 3, 245.\
> > > > > [3] Kriegeskorte, N., Mur, M., & Bandettini, P. A. (2008). Representational similarity analysis-connecting the branches of systems neuroscience. Frontiers in Systems Neuroscience, 4.

---

> > > > > > ### Comment · Reviewer_u84n · 2022-11-25
> > > > > > **Response**
> > > > > >
> > > > > > Thanks for all of the work. I have raised my score. Great job on this paper.
> > > > > >
> > > > > > One final ask: I believe you should cite the paper below, which is going to be published in NeurIPS 2022 and is highly relevant. It also introduces a method that might be appropriate for fixing the observed misalignment.
> > > > > >
> > > > > > Fel T, Flipe I, Linsley D, Serre T. Harmonizing the object recognition strategies of deep neural networks with humans. NeurIPS 2022.

---

> > > > > > > ### Author Response · Authors · 2022-11-25
> > > > > > > **Author response**
> > > > > > >
> > > > > > > Thanks for the pointer! This sounds interesting. We will add this reference to the final version of our manuscript.

---

> > > ### Comment · Reviewer_u84n · 2022-11-17
> > > **Response**
> > >
> > > > We are not sure what else is going on in the “varying architecture” plot aside from varying the architecture. All of the models in Kornblith et al. (2019) [4] were trained with the same objective function (softmax cross-entropy error) and the same hyperparameters (learning rate, optimizer, regularization, etc.). The only variable that differs between the points in the “varying architecture” plot is the architecture. Given that varying the entire architecture, including width and depth, has no meaningful impact on odd-one-out accuracy, we do not see how varying only width or depth could lead to different conclusions.
> > >
> > > Sorry if I'm being dense here, but each of those architectures have different mechanisms (e.g., skips vs. transformers), widths, depths, normalizations, etc. My proposal was to pick one and then varying depth/width in that architecture. A null result for current fig 3 does not guarantee a null result for the controlled experiment I proposed. Also, I want to say, I don't expect that to make much of a difference — I agree that for THINGS language grounding is probably most important — but I don't think the current experiments are the right controls (although I may be misunderstanding).
> > >
> > > > Nonetheless, we agree with the reviewer that this finding is not a core finding of our study and we have reduced its emphasis.
> > >
> > > Great.
> > >
> > > > Thanks for pointing this out.
> > >
> > > Great.

---

> > > > ### Author Response · Authors · 2022-11-25
> > > > **Author response**
> > > >
> > > > > Sorry if I'm being dense here, but each of those architectures have different mechanisms (e.g., skips vs. transformers), widths, depths, normalizations, etc. My proposal was to pick one and then varying depth/width in that architecture. A null result for current fig 3 does not guarantee a null result for the controlled experiment I proposed. Also, I want to say, I don't expect that to make much of a difference — I agree that for THINGS language grounding is probably most important — but I don't think the current experiments are the right controls (although I may be misunderstanding).
> > > >
> > > > Although we see where you are coming from, we are convinced that null results for our experiments will give a null result for the experiments you propose. Figure 3 (top right) contains 3 ResNets and 3 DenseNets that vary only in depth and a pair of MobileNets that vary only in width. We see no meaningful differences among these sets of networks, nor between the different families of architectures shown in this plot, which vary in width, depth, kernel size, and connectivity between layers but are otherwise trained with identical settings. If varying all of these factors does not change the strength of alignment, why would varying only one of them lead to a difference in alignment? We agree that controlled experiments are useful for finding sources of variability in results between networks, but in this case, we see no variability that needs to be causally attributed.

---

> > ### Comment · Reviewer_u84n · 2022-11-17
> > **Response**
> >
> > > Strong performance on computer vision tasks does not necessarily mean that a model has learned a concept space that is aligned with a human’s concept space.
> >
> > OK fair point. Where my comment came from is presenting this finding on THINGS in isolation raises questions about what it means and reflects about similarities/differences in DNN vs. human vision. Does the improved ability of CLIP vs. purely imagenet-trained models to align with humans reflect the former's access to semantic information that humans relied on for rendering their decisions? What alignment is being tested? Those types of computational insights are what I was hoping for here.
> >
> > > ... it does not seem straightforward to improve human alignment by modifying the architecture, whereas increasing the diversity of the
> >
> > training data or changing the training objective appears to make a difference. This could help guide machine learning researchers if they are interested in developing models whose representation spaces are aligned with human similarity judgments.
> > Good point.
> >
> > > ... Scaling laws from previous work relate accuracy to both model size and dataset size...
> >
> > Good point.
> > > ... human representations that neural network representations currently lack, which may eventually lead to better task performance...
> >
> > In the absence of evidence for this, I advise seeking another benchmark where you can show improving alignment on THINGS improves performance there. This would have the additional benefit of offering computational insights into what is missing with DNNs vs. humans.

---

> > > ### Author Response · Authors · 2022-11-25
> > > **Author response**
> > >
> > > > OK fair point. Where my comment came from is presenting this finding on THINGS in isolation raises questions about what it means and reflects about similarities/differences in DNN vs. human vision. Does the improved ability of CLIP vs. purely imagenet-trained models to align with humans reflect the former's access to semantic information that humans relied on for rendering their decisions? What alignment is being tested? Those types of computational insights are what I was hoping for here.
> > >
> > > Although the THINGS triplet odd-one-out task data is the largest collection of human similarity judgments for natural object images we are aware of, we agree that results for an additional human similarity judgment dataset would corroborate our findings and help to interpret results.
> > >
> > > Thus, we requested permission to use the similarity judgments obtained from the multi-arrangement task used in Cichy et al. (2019) [1] and performed the same set of analyses that we have performed for THINGS for their data. We find highly similar results for the similarity judgments from Cichy et al. (2019).
> > >
> > > - Image/text models are substantially better aligned with human similarity judgments than ImageNet models. (Link:  https://i.imgur.com/xLNuODc.png)
> > > - Varying the objective function in ImageNet models impacts their alignment, whereas changing the architecture seems to have no impact on alignment whatsoever. (Link: https://i.imgur.com/SmppCpU.png )
> > > - SSL models trained with a contrastive learning objective (e.g., SimCLR, MoCov) are better aligned than both non-Siamese (e.g., Rotnet, Jigsaw) and non-contrastive models (e.g., BarlowTwins). (Link: https://i.imgur.com/jMCqJNT.png )
> > > - Scaling ImageNet models does not improve human alignment. (Link: https://i.imgur.com/jMCqJNT.png )
> > >
> > > The multi-arrangement task is another commonly used task in the cognitive sciences to measure (human) similarity judgments [2]. To perform this task, subjects arrange images on a computer screen so that the distances between them reflect their similarities. To measure alignment between human and neural net representation spaces, one typically performs representational similarity analysis (RSA) [3] on the representation space and computes correlation coefficients with the human similarity judgments rather than measuring accuracy [1]. Given that the multi-arrangement task and analysis strategy are quite different from the THINGS triplet odd-one-out task but our results remain consistent, we are confident that our findings reflect true properties of human/machine alignment rather than idiosyncrasies of the task.
> > >
> > > > In the absence of evidence for this, I advise seeking another benchmark where you can show improving alignment on THINGS improves performance there. This would have the additional benefit of offering computational insights into what is missing with DNNs vs. humans.
> > >
> > > In addition to the above evaluations, we examined whether the transformation matrix obtained from linear probing on the THINGS triplet odd-one-out task transfers to the multi-arrangement task and improves alignment with similarity judgments. We find that our probe transfers to the other data and gives improved alignment compared to not using the probe at all (Link: https://i.imgur.com/B52Awi7.png ). We believe that this yields insights that were missing before (i.e., DNN representation spaces do not properly weigh the features that humans use when performing similarity judgments about objects in the world).

---

### Official Review · Reviewer_hcki · 2022-10-25

**Confidence:** 3
**Correctness:** 1
**Technical Novelty And Significance:** 3
**Empirical Novelty And Significance:** Not applicable
**Recommendation:** 1

**Clarity, Quality, Novelty And Reproducibility:**

Despite the clear flow. There are some ambiguities that obstructs the understanding:  the consistency among users were not provided: is it 67.22 (sec 3)?



**Strength And Weaknesses:**

This paper is in good flow so that I can easily follow.
The direction of research is inherently interesting and could be of great importance for many researches such as network interpretability.
The experiment design is comprehensive. I think this would be a very solid paper if I had agreed with the methodologies provided by the paper (see the reason below).


**Summary Of The Paper:**

This paper explores the how human's representation of visual images is aligned with network representations.
For this goal, this paper utilized an "odd-one-out" test setting. This paper built a sparse Bayesian model to represent human's responses. And utilizes multiple methods such as zero-shot, linear probing and regression. By comparing the choices from the machine features under these distance metrics to human's choices. This paper identified factors in deep models that affect the alignments.


**Summary Of The Review:**

I fully respect the effort in this paper and I think all the experiments seem solid.
However, I strongly disagree with the method utilized by this paper to measure the alignment between the two representations.
I believe one acceptable way would be measuring both representations based on separate signal sources (human representation from EEG etc), then analyze  the correlation. And due to the complexity, human representation should not be only based on one task.
We are still not able to build up a real human representation. The models proposed by the paper are based on representations of human behavior (decisions). This representation, however, is only based on observations, not based on cog signals. And even for the representation of human decisions, there can be many different representations, Why is the odd-on-out method the right way?
In addition, as also pointed out by the paper in the discussion section, the conclusion on machine/human feature similarity is heavily dependent on the metrics used. With a slight different metric, there are can be totally different conclusions.

---

> ### Author Response · Authors · 2022-11-11
> **Author reponse (1/2)**
>
> **I strongly disagree with the method utilized by this paper to measure the alignment between the two representations. I believe one acceptable way would be measuring both representations based on separate signal sources (human representation from EEG etc), then analyze the correlation. And due to the complexity, human representation should not be only based on one task. We are still not able to build up a real human representation. The models proposed by the paper are based on representations of human behavior (decisions). This representation, however, is only based on observations, not based on cog signals.**
>
> The study of inferred human mental representations of objects as a surrogate for how the human mind (visually and semantically) represents objects dates back to at least the 1950s. The existence of these mental representations was core to David Marr’s work in the 1970s on the different levels of analysis of an information processing system [1,2], where the algorithmic level is considered the representational level. (This work inspired David Rumelhart, James McClelland, and Geoffrey Hinton to investigate Parallel Distributed Processing models [3,4] which are the foundation for modern neural nets.) Following Marr, a large body of work in cognitive science has sought to infer properties of mental representations from human behavior and has acknowledged the importance of the algorithmic/representational level for understanding how the human mind organizes/distributes knowledge about the (visual) world (e.g, [5-8] to name a few). The study of the properties of the mental representations that give rise to human behavioral similarity judgments dates back to the early work of Amos Tversky [9] and has been discussed and agreed upon in many succeeding works (e.g., [8;10-14]).
>
> Given this long history of cognitive science research that presumes a) the existence of mental representations of objects which can b) be derived from (human) behavior through similarity judgments, and c) whose study is crucial to understand how the (human) mind organizes knowledge, we remain convinced that we have pursued a scientifically valid path.
>
> In addition, we want to remark that our goal has never been to make any claims about predicting brain signals from neural network representations. We compared neural net representations with mental representations inferred from human behavior. Moreover, measurements of brain activity of the type suggested by the reviewer are *not* “cog signals”—the fields of cognitive psychology and cognitive science specifically study mental representations measured through behavior, as opposed to representations measured directly from the brain [15,16,20].
>
> In summary, by dismissing the idea that properties of mental representations can be ascertained from behavior, the reviewer dismisses not only the validity of our work but an enormous amount of previous work in cognitive science. Although the reviewer is entitled to their opinion, we do not believe that it is fair to reject our work on this basis.
>
> **And even for the representation of human decisions, there can be many different representations, Why is the odd-on-out method the right way?**
>
> We want to clarify that we don’t claim that the triplet odd-one-out task is the best way to measure alignment. However, given its frequent use in metric learning and the cognitive sciences, the triplet odd-one-out task appears to be a reasonable way to measure the alignment between metric spaces of (pretrained) neural nets and spaces humans have learned for performing pairwise similarity judgments. For a more detailed explanation of why the triplet odd-one-out task is generally a useful task for deriving mental representations and may be preferred over numerical ratings, please see our response to reviewer fmAV.
>
> **In addition, as also pointed out by the paper in the discussion section, the conclusion on machine/human feature similarity is heavily dependent on the metrics used. With a slight different metric, there are can be totally different conclusions.**
>
> Whenever we measure the performance of ML algorithms, there are choices about how performance is measured that can affect results. The best that one can do is to investigate several metrics and show that results are consistent across them. Here, we investigate zero-shot, linear probing, and regression performances while varying the similarity kernel (dot product vs. cosine similarity) and find that the results are highly consistent across the different setups. It is always possible that a different metric would give completely different rankings, but we do not believe that this is likely for sane choices of metrics.
>
>
> **There are some ambiguities that obstructs the understanding: the consistency among users were not provided: is it 67.22 (sec 3)?**
>
> Yes, the ceiling accuracy based on consistency across subjects is 67.22%. For more details about how this number was computed, see [16] or [21].

---

> > ### Author Response · Authors · 2022-11-11
> > **Author response (2/2)**
> >
> > **References**
> >
> > [1] Marr, D., & Poggio, T. (1976). From understanding computation to understanding neural circuitry.
> >
> > [2] Marr, D., & Nishihara, H. K. (1978). Representation and recognition of three-dimensional shapes. Proceedings of the Royal Society of London. Series B. 200. 269-294.
> >
> > [3] Rumelhart, D. E., Smolensky, P., McClelland, J. L., & Hinton, G. (1986). Sequential thought processes in PDP models. Parallel distributed processing: explorations in the microstructures of cognition, 2, 3-57.
> >
> > [4] Rumelhart, D. E., McClelland, J. L., & PDP Research Group. (1988). Parallel distributed processing (Vol. 1, pp. 354-362). New York: IEEE.
> >
> > [5] Rosch, E., Mervis, C. B., Gray, W., Johnson, D., & Boyes-Braem, P. (1976). Basic objects in natural categories. Cognitive Psychology, 8, 382-439.
> >
> > [6] Fodor, J. A. (1983). Representations: Philosophical essays on the foundations of cognitive science. Cambridge, MA: MIT Press.
> >
> > [7] Biederman, I. (1987). Recognition-by-components: a theory of human image understanding. Psychological Review, 94(2), 115.
> >
> > [8] Edelman, S. (1998). Representation is representation of similarities. Behavioral and Brain Sciences, 21(4), 449-467.
> >
> > [9] Tversky, A. (1977). Features of similarity. Psychological Review, 84(4), 327.
> >
> > [10] Nosofsky, R. M. (1986). Attention, similarity, and the identification–categorization relationship. Journal of experimental psychology: General, 115(1), 39.
> >
> > [11] Medin, D. L. (1989). Concepts and conceptual structure. American Psychologist, 44(12), 1469.
> >
> > [12] Goldstone, R. L. (1994). The role of similarity in categorization: Providing a groundwork. Cognition, 52(2), 125-157.
> >
> > [13] Hahn, U., & Chater, N. (1997). Concepts and similarity. Knowledge, concepts and categories, 43-92.
> >
> > [14] Goldstone, R. L., & Son, J. Y. (2012). Similarity. Oxford University Press.
> >
> > [15] Kriegeskorte, N., & Kievit, R. A. (2013). Representational geometry: integrating cognition, computation, and the brain. Trends in cognitive sciences, 17(8), 401-412.
> >
> > [16] Hebart, M. N., Zheng, C. Y., Pereira, F., & Baker, C. I. (2020). Revealing the multidimensional mental representations of natural objects underlying human similarity judgments. Nature Human Behaviour, 4(11), 1173-1185.
> >
> > [17] Torgerson, W. S. (1954). Multidimensional Scaling I: Theory and Method. Psychometrika, 17(4), 401-419.
> >
> > [18] Fukuzawa, K., Itoh, M., Sasanuma, S., Suzuki, T., Fukusako, Y., & Masui, T. (1988). Internal representations and the conceptual operation of color in pure alexia with color naming defects. Brain and Language, 34(1), 98-126.
> >
> > [19] Robilotto, R., & Zaidi, Q. (2004). Limits of lightness identification for real objects under natural viewing conditions. Journal of Vision, 4(9), 9-9
> >
> > [20] Zheng C. Y., Pereira F., Baker C. I., & Hebart M. N. (2019). Revealing interpretable object representations from human behavior. In 7th International Conference on Learning Representations, ICLR 2019.
> >
> > [21] Muttenthaler, L., Zheng C. Y., McClure P., Vandermeulen R. A., Hebart M. N., &, Pereira F. (2022). VICE: Variational Interpretable Concept Embeddings. Advances in Neural
> > Information Processing Systems 36: Annual Conference on Neural Information Processing
> > Systems 2022, NeurIPS 2022.

---

> ### Comment · Reviewer_u84n · 2022-11-13
> **Statement of concern**
>
> > However, I strongly disagree with the method utilized by this paper to measure the alignment between the two representations. I believe one acceptable way would be measuring both representations based on separate signal sources (human representation from EEG etc), then analyze the correlation. And due to the complexity, human representation should not be only based on one task. We are still not able to build up a real human representation. The models proposed by the paper are based on representations of human behavior (decisions). This representation, however, is only based on observations, not based on cog signals. And even for the representation of human decisions, there can be many different representations, Why is the odd-on-out method the right way?
>
> After reading through the authors’ responses to my critiques, I checked through to see what the other reviewers thought. I hate having to do this but I wanted to raise concerns about this review. This section contains the main critiques, but they make no sense and convey a lack of background in cognitive science or AI/human comparative work. I hope that the AC and the other reviewers do not weigh this review into their decisions.

---

### Official Review · Reviewer_fmAV · 2022-10-25

**Confidence:** 3
**Correctness:** 4
**Technical Novelty And Significance:** 3
**Empirical Novelty And Significance:** 3
**Recommendation:** 6

**Clarity, Quality, Novelty And Reproducibility:**

Clarity/Quality: Strong

Novelty: There has been a lot of work on learning "human-compatible representations" for specific tasks and downstream settings, so I think the main novelty of this paper is not the consideration of whether representations are aligned with human perceptual representations, but rather whether properties such as scale affects them. I largely consider this limited novelty, as its focused on a very specific measure of human alignment.

Reproducibility: The THINGS dataset actually used seems to be different from the examples the authors are able toe show, but barring that everything else seems reproducible.


**Strength And Weaknesses:**

S1: Paper is very clear and easy to follow.

S2. The paper conducts a very systematic and thorough investigation across a variety of different models architectures and settings, thus strongly supporting their claim that scale does not result in improved human alignment.


W1: Why is the odd-on-out method the right way to measure human alignment of representations (versus pair similarities as in Peterson (2018) or other approaches)? Given that the paper's primary goal is to make a thorough claim on what properties (e.g. scale, objective) affect the learned representations alignment with humans, there is currently not much discussion on why the triple task is the best way to measure alignment.

W2 (minor): The reference triplet task for CIFAR-100 appears quite different in spirit from the triplet task with human responses on the THINGS dataset, so I'm not sure how useful it is as a baseline to compare with. The process itself is explicitly "coarse" class driven, unlike the human judgements which aren't necessarily driven by class/object.



**Summary Of The Paper:**

The paper studies the degree representations of computer vision models are aligned with human judgements of similarity, and find that changes in model size and architecture have little effect on the degree of a model's alignment, while training data does. The authors show this result via a thorough set of experiments comparing many models and settings, measuring human aligning via annotations from an odd-one-out task.

**Summary Of The Review:**

I think the paper is a great contribution to the ICLR community, both from its thoroughness in experiments and its timeliness as we seek to understand the full benefits of large, pre-trained models and the role of pre-training data. My primary concern is the assumption that the triplet task is fully representative of "human alignment" as a general property, which can be addressed with more discussion, and perhaps examples from the annotation task, with which I'm happy to increase my score.

---

> ### Author Response · Authors · 2022-11-11
> **Author response (1/2)**
>
> **Why is the odd-on-out method the right way to measure human alignment of representations (versus pair similarities as in Peterson (2018) or other approaches)? Given that the paper's primary goal is to make a thorough claim on what properties (e.g. scale, objective) affect the learned representations alignment with humans, there is currently not much discussion on why the triple task is the best way to measure alignment.**
>
> We acknowledge that we have not thoroughly discussed why the triplet odd-one-out task is a useful method to measure alignment between human and neural network representation spaces.
> We use the odd-one-out task because it has been adopted in previous work to probe humans’ intuition about object similarities without introducing bias and because a large dataset of human odd-one-out judgments is available [1-3].  An alternative approach would be to ask participants for numerical ratings of similarity (e.g. on a scale of 0-10) for pairs of objects, as in Peterson et al. (2018), instead of triplets. However, participants may differ in how they calibrate their rating scales and are often inconsistent. Two participants with potentially the same notion of object similarity might give quite different numerical ratings, because each person's way of converting their intuitive sense of similarity to a numerical rating may be accomplished through an individually unique and somewhat arbitrary mapping. Moreover, when objects are not closely related, it may be difficult for a subject to determine what numerical similarity score to assign to them. In contrast, two participants with the same internal representations of object similarities should be able to agree with regard to which object is the least similar to the other two [4].
>
> The triplet task also makes fewer assumptions about the relationship between human and neural network representations. For similarities in a neural network representation space to exactly match numerical human similarity judgments, human and machine representation spaces would need to be identical up to a rotation, whereas maximal triplet odd-one-out accuracy can be achieved even if one space is slightly warped with respect to the other. In Machine Learning, the triplet task has been widely used as an objective function in metric learning, where available supervision signals rarely provide a ground-truth similarity measure but are often sufficient to infer that one image is more similar to a second than a third (e.g., [5-7]). We have now included a shortened version of the above discussion under a separate “Triplet Task” subsection in the Methods section of the main text.
>
>
> We want to clarify that we don’t claim that the triplet odd-one-out task is the best way to measure alignment. However, given its frequent use in metric learning and the cognitive sciences, the triplet odd-one-out task appears to be a reasonable way to measure the alignment between metric spaces of (pretrained) neural nets and spaces humans have learned for performing pairwise similarity judgments.
>
> **The reference triplet task for CIFAR-100 appears quite different in spirit from the triplet task with human responses on the THINGS dataset, so I'm not sure how useful it is as a baseline to compare with. The process itself is explicitly "coarse" class driven, unlike the human judgements which aren't necessarily driven by class/object.**
>
> We agree with the reviewer that the CIFAR-100 coarse triplet task is different from the THINGS triplet odd-one-out task. The main reason for constructing the CIFAR-100 coarse triplet task was to examine whether or not any findings from comparing human to neural network responses for the THINGS triplet odd-one-out task can be attributed to the nature of the triplet task. The CIFAR-100 coarse task is somewhat comparable to the THINGS task because both tasks can be performed given a relatively small set of concepts, as Hebart et al. (2020) and Muttenthaler et al. (2022) have previously shown for the THINGS task. Because ImageNet accuracy correlates well with odd-one-out accuracy on the CIFAR-100 coarse task, we can rule out the possibility that the reason that ImageNet accuracy correlates poorly with odd-one-out accuracy on the THINGS task is attributable to the nature of the triplet task itself. We have added a more detailed explanation of our rationale for the CIFAR-100 triplet task to Appendix C which we hope will clarify any confusion about this task.

---

> > ### Author Response · Authors · 2022-11-11
> > **Author response (2/2)**
> >
> > **Reproducibility: The THINGS dataset actually used seems to be different from the examples the authors are able to show, …**
> >
> > For evaluating neural networks on the THINGS triplet odd-one-out task, we used the same images that were used for collecting human responses in Hebart et al. (2020). Hence, our results are fully reproducible with the original THINGS dataset. Since those images are unfortunately not copyright-free, in the manuscript we show copyright-free images that were collected by the same authors as the original THINGS data [8] instead of the original images.
> >
> > **Final Note**
> >
> > We thank the reviewer for their overall positive feedback and their suggestion on discussing the triplet odd-one-out task as a useful method to measure alignment between human and neural network representations. We have added a shortened version of the above discussion on the odd-one-out task to the main text of our manuscript (see Methods) as well as a detailed explanation of our rationale for the CIFAR-100 triplet task (see Appendix C).
> >
> > **References**
> >
> > [1] Hebart, M. N., Zheng, C. Y., Pereira, F., & Baker, C. I. (2020). Revealing the multidimensional mental representations of natural objects underlying human similarity judgements. Nature Human Behaviour, 4(11), 1173-1185.
> >
> > [2] Charles Y. Zheng, Francisco Pereira, Chris I. Baker, and Martin N. Hebart. Revealing interpretable object representations from human behavior. In 7th International Conference on Learning Representations, ICLR 2019.
> >
> > [3] Lukas Muttenthaler, Charles Y. Zheng, Patrick McClure, Martin N. Hebart, and  Francisco Pereira. VICE: Variational Interpretable Concept Embeddings. Advances in Neural Information Processing Systems 36: Annual Conference on Neural Information Processing
> > Systems 2022, NeurIPS 2022.
> >
> > [4] Maddox, W. T., & Ashby, F. G. (1993). Comparing decision bound and exemplar models of categorization. Perception & psychophysics, 53(1), 49-70.
> >
> > [5] ​​Chechik, G., Sharma, V., Shalit, U., & Bengio, S. (2010). Large Scale Online Learning of Image Similarity Through Ranking. Journal of Machine Learning Research, 11(3).
> >
> > [6] Wang, X., & Gupta, A. (2015). Unsupervised learning of visual representations using videos. In Proceedings of the IEEE international conference on computer vision (pp. 2794-2802).
> >
> > [7] Amid, E. & Ukkonen, A.. (2015). Multiview Triplet Embedding: Learning Attributes in Multiple Maps. Proceedings of the 32nd International Conference on Machine Learning, in Proceedings of Machine Learning Research 37:1472-1480.
> >
> > [8] Stoinski, L. M., Perkuhn, J., & Hebart, M. N. (2022). THINGS+: New Norms and Metadata for the THINGS Database of 1,854 Object Concepts and 26,107 Natural Object Images.

---

### Author Response · Authors · 2022-11-16
**Reminder: first stage of the discussion period ends on November 18th**

Dear reviewers fmAV, hcki, and u84n,

We want to reach out again to check if there are any additional questions or concerns about our rebuttal that we can address before the first stage of the discussion period ends.

Thanks again for taking the time to read our work and providing helpful feedback!

Paper2842 Authors

---

### Author Response · Authors · 2022-11-25
**General response (1/2)**

We would like to thank all reviewers for taking the time to provide thoughtful reviews and help us to improve our manuscript. We were pleased to see that the reviewers fmAV and u84n recommended the paper for acceptance and found many positive aspects to our paper.

**fmAV**:
- “The paper is very clear and easy to follow.”
- “I think the paper is a great contribution to the ICLR community, both from its thoroughness in experiments and its timeliness as we seek to understand the full benefits of large, pre-trained models and the role of pre-training data.”

**u84n**:
- “Awesome introduction.”
 - “... it is indeed intriguing and well-done work that may signify a new direction in the field of comp cog neuro towards embracing and investigating the scaling laws which are driving performance in AI.”

Here we address a couple of concerns that were shared among more than one reviewer. We have also provided individual reviewer responses.

**-Additional analysis beyond THINGS-**

We requested permission to use the similarity judgments obtained from the multi-arrangement task used in Cichy et al. (2019) [1] and performed the same set of analyses that we have performed for THINGS for their human similarity data. The multi-arrangement task is another commonly used task in the cognitive sciences to measure (human) similarity judgments [2]. To measure alignment between human and neural net representation spaces, one typically performs representational similarity analysis (RSA) [3] and computes Spearman correlation coefficients rather than measuring accuracy. The multi-arrangement task is quite different from the triplet odd-one-out task and thus by performing our analyses on this data we are able to verify the generalizability of our results.
Acquiring this data and performing analyses took longer than the first stage of the discussion period, but we are happy to share the results with you now. We have confirmed with the PC that we are allowed to share links showing the additional results. We find highly similar, if not identical, results for the similarity judgments from Cichy et al. (2019),

- Image/text models are substantially better aligned with human similarity judgments than ImageNet models. (Link:  https://i.imgur.com/xLNuODc.png)
- Varying the objective function in ImageNet models impacts their alignment, whereas changing the architecture seems to have no impact on alignment whatsoever. (Link: https://i.imgur.com/SmppCpU.png )
- SSL models trained with a contrastive learning objective (e.g., SimCLR, MoCov) are better aligned than both non-Siamese (e.g., Rotnet, Jigsaw) and non-contrastive models (e.g., BarlowTwins). (Link: https://i.imgur.com/jMCqJNT.png )
- Scaling ImageNet models does not improve human alignment. (Link: https://i.imgur.com/jMCqJNT.png )
- Furthermore, we find that the transformation matrix obtained from linear probing on the things triplet odd-one-out task transfers to the multi-arrangement task and gives improved alignment compared to using no additional transform. (Link: https://i.imgur.com/B52Awi7.png )

We will add all findings from the additional analysis to the camera-ready version of our manuscript. For now, reviewers can find plots showing those results under the anonymously shared links adjacent to each bullet point.

**-Triplet odd-one-out task-**

Although we believe the additional analyses above should address Reviewer fmAV’s concerns regarding “the assumption that the triplet task is fully representative of human alignment,” we have also added additional discussion of the triplet task as the reviewer suggests. In particular, we now describe the advantages of this task over others in a paragraph in the main text (see revised manuscript):

“Hebart et al. (2020) collected similarity judgments from human participants on images in THINGS in the form of responses to a triplet task. In this task, images from three distinct categories are presented to a participant, and the participant selects the image that is most different from the other two (or equivalently the pair that are most similar). The triplet task has been used to study properties of human mental representation for many decades (e.g., [4,5,6]), as it measures humans' intuitions regarding object similarity without introducing bias. Compared to tasks involving numerical/Likert-scale pairwise similarity judgments, the triplet task does not require different subjects to interpret the scale similarly, and does not require that the degree of perceived similarity is cognitively accessible.”

**Final note**

We want to thank everyone for taking the time to read our paper and providing useful comments.

---

> ### Author Response · Authors · 2022-11-25
> **General response (2/2)**
>
> **References**
>
>
> [1] Cichy, R. M., Kriegeskorte, N., Jozwik, K. M., van den Bosch, J. J., & Charest, I. (2019). The spatiotemporal neural dynamics underlying perceived similarity for real-world objects. Neuroimage, 194, 12-24.\
> [2] Kriegeskorte, N., & Mur, M. (2012). Inverse MDS: Inferring dissimilarity structure from multiple item arrangements. Frontiers in Psychology, 3, 245.\
> [3] Kriegeskorte, N., Mur, M., & Bandettini, P. A. (2008). Representational similarity analysis-connecting the branches of systems neuroscience. Frontiers in Systems Neuroscience, 4.\
> [4] Fukuzawa, K., Itoh, M., Sasanuma, S., Suzuki, T., Fukusako, Y., & Masui, T. (1988). Internal representations and the conceptual operation of color in pure alexia with color naming defects. Brain and Language, 34(1), 98-126.\
> [5] Robilotto, R., & Zaidi, Q. (2004). Limits of lightness identification for real objects under natural viewing conditions. Journal of Vision, 4(9), 9-9.\
> [6] Hebart, M. N., Zheng, C. Y., Pereira, F., & Baker, C. I. (2020). Revealing the multidimensional mental representations of natural objects underlying human similarity judgements. Nature human behaviour, 4(11), 1173-1185.

---

### Comment · Area_Chair_jC6M · 2022-12-03
**Requesting Author Response**

Dear authors,

The analysis and conclusions of this paper have major flaws (explained below). Since these issues were not brought up in the initial reviews, I would like to give you a chance to respond to these comments before we make the final decision.

- The dataset used for the analysis has 1854 categories, while most of the models in the paper are trained on ImageNet, which has only 334 classes in common with that dataset (less than 20%). So, the ImageNet-based models have not seen about 80% of the categories during training. Comparing humans with a model that has not seen 80% of those objects is not fair. Basically, the paper compares in-distribution evaluation with out-of-distribution evaluation.

- The maximum achievable odd-one-out-accuracy is about 67%. That means even humans cannot do the task correctly about 33% of the time. Therefore, most of the rankings and conclusions are not reliable due to this large error margin.

- Comparing CLIP models with ImageNet-based models is unfair. CLIP models have seen many more categories, and there might be a higher overlap between the categories in the Things dataset. The better performance of CLIP should not be attributed to language/text. They are trained on data that includes more categories of the Things dataset.

- It is shown in the literature that models that achieve higher accuracy on ImageNet are not necessarily better at other tasks since they are over-optimized for ImageNet. So, conclusions such as “contrasting positive against negative examples rather than solely using positive examples improves alignment with human similarity judgments.” are not necessarily correct. The models in the paper are used for evaluation of out-of-distribution generalization (generalization to 1520 unseen categories). So, it is expected that a model highly optimized for ImageNet does not generalize well.

Thanks,

Area Chair

---

> ### Author Response · Authors · 2022-12-06
> **Author response (1/4)**
>
> Dear Area Chair,
>
> Thank you for taking the time to read our paper and provide your feedback. We address each of your points below.
>
> > The dataset used for the analysis has 1854 categories, while most of the models in the paper are trained on ImageNet, which has only 334 classes in common with that dataset (less than 20%). So, the ImageNet-based models have not seen about 80% of the categories during training. Comparing humans with a model that has not seen 80% of those objects is not fair. Basically, the paper compares in-distribution evaluation with out-of-distribution evaluation.
>
> Although it would be interesting to explore the extent to which category overlap between the pretraining dataset and the images for which human similarity judgments were obtained affects odd-one-out accuracy, we do not believe that such an investigation is likely to change our findings.
>
> First, being able to classify the categories to which the 1854 images from THINGS belong is neither necessary nor sufficient to learn a representation that matches human similarity judgments. For example, ImageNet contains many animals, but ImageNet-trained models still struggle with the animal concept. Conversely, a model could learn the animal concept without learning how to classify particular animal species. In addition, some concepts found by VICE, which directly models human odd-one-out responses on THINGS, do not align with object categories at all, e.g. “colorful” (dimension 14) or “patterns” (dimension 15). ImageNet models nevertheless learn sufficiently rich representations to capture these dimensions, as we shown in Figure J.6 of the appendix (p. 34).
>
> Second, related to the point above, THINGS and ImageNet are both datasets of natural images of objects. Thus, we analyze the representation spaces of models trained on natural objects for a task that uses another natural image database rather than for assessing their representations on drawings, sketches, or medical data of which the latter set clearly is out-of-distribution. Our setting is comparable to well-explored transfer settings where representations from ImageNet-pretrained models have been found to be good for downstream tasks (cf. Donahue et al., 2014; Razavian et al., 2014).
>
> Third, THINGS and ImageNet have nontrivial overlap. The 334 classes that the two image datasets share are *exact* matches between the corresponding WordNet IDs. There actually is a substantially larger overlap of categories between the two datasets (Hebart et al., 2019; Hebart et al. 2020). 802 of the ImageNet-1K classes are either a THINGS category or have a superclass in THINGS (e.g., “dog” is a superset of a specific dog breed). We will elaborate on the difference between exact WordNet ID matches and this less conservative computation of the class intersection between the two datasets in more detail in the final version of our manuscript. However, this does not change any of the conclusions about our findings.
>
> Finally, during the course of the rebuttal, we additionally evaluated the representation space of each model on another dataset of human similarity judgments where humans performed a multi-arrangement task from Cichy et al. (2019). The images from this multi-arrangement task are from ImageNet, so concerns regarding dataset distribution do not apply. Nevertheless, we observe the same findings as we’ve observed for the THINGS dataset. We provide links to these results in a comment below this one.

---

> > ### Author Response · Authors · 2022-12-06
> > **Author response (2/4)**
> >
> > > The maximum achievable odd-one-out-accuracy is about 67%. That means even humans cannot do the task correctly about 33% of the time. Therefore, most of the rankings and conclusions are not reliable due to this large error margin.
> >
> > We suspect that there is a misunderstanding. The goal of this paper was *not* to examine whether humans or neural networks perform better on the triplet odd-one-out task. There are no correct answers for the triplet odd-one-out task. The triplet odd-one-out task is a means for collecting human similarity judgments and understanding the concept space according to which humans make odd-one-out choices in sets of three. Through repeated samplings of a subset of triplets, Hebart et al. (2020) found that some triplets had more consistent responses than others, which ultimately yielded a ceiling accuracy of 67% for predicting human responses to a randomly selected triplet. We elaborate on this in great detail both in the manuscript and in the rebuttal. Both Zheng et al. (2019), Hebart et al. (2020), and Muttenthaler et al. (2022) show this in their analyses of the human concept spaces inferred from the triplet choices we use for our experiments.
> >
> > The existence of a ceiling accuracy level is common in studies that aim to measure alignment between machine outputs and behavioral/brain data rather than some objectively measurable ground truth (e.g., Yamins et al., 2014; Peterson et al., 2018), and does not imply that our results are unreliable. The ceiling accuracy level is irrelevant to the determination of whether a difference between two accuracy numbers is statistically significant or not. Due to the large number of triplets available for evaluation, even small differences in test accuracy are statistically significant. For the experiment that uses the fewest triplets for evaluating models — the linear probing experiments with only 54k test samples during cross-validation (see Appendix A.1) — an accuracy gap of $0.6\%$ is statistically significant with $\alpha = 0.05$ using a two-sided two-sample Z-test. For the zero-shot experiments with more than one million triplets, a difference of $0.1\%$ is statistically significant.
> >
> >
> > > Comparing CLIP models with ImageNet-based models is unfair. CLIP models have seen many more categories, and there might be a higher overlap between the categories in the Things dataset. The better performance of CLIP should not be attributed to language/text. They are trained on data that includes more categories of the Things dataset.
> >
> > We do not intend to claim that “... better performance of CLIP should ... be attributed to language/text.” We don’t know whether the higher degree of alignment of image/text models is attributable to their multimodal training. To the contrary, we observe that the ViT-G/14 JFT model, which was trained on a classification task much larger than ImageNet, performs on par with the image/text models, suggesting that dataset diversity (potentially including the number of classes), and not the use of text, is not a critical factor. We will clarify this.
> >
> > However, given the analysis of the Cichy et al. multi-arrangement task we have performed during the response period, it is unlikely that the difference between image/text models and ImageNet models can be attributed solely or primarily to the “fact” that their training data includes greater overlap with THINGS (this is very plausible, but not entirely assured; OpenAI released few details of the CLIP dataset collection process, and we are not aware of any source showing that the CLIP data shares more categories with THINGS than ImageNet). All of the 118 images used for the multi-arrangement task in Cichy et al. (2019) are from the ImageNet database. So if anything, ImageNet models have an advantage here. However, we find the same results for this data as we’ve found for the triplet data (see general response and response to reviewer u84n).
> >
> > We recognize that our work only begins to answer the question of how different supervision signals affect human alignment and we would be happy to discuss the open questions that remain further in the camera ready if the paper is accepted. However, we also do not believe that there is a trivial explanation for the superiority of image/text models and models trained on larger image classification datasets that we have missed.

---

> > > ### Author Response · Authors · 2022-12-06
> > > **Author response (3/4)**
> > >
> > > > It is shown in the literature that models that achieve higher accuracy on ImageNet are not necessarily better at other tasks since they are over-optimized for ImageNet. So, conclusions such as “contrasting positive against negative examples rather than solely using positive examples improves alignment with human similarity judgments.” are not necessarily correct. The models in the paper are used for evaluation of out-of-distribution generalization (generalization to 1520 unseen categories). So, it is expected that a model highly optimized for ImageNet does not generalize well.
> > >
> > > We are not sure exactly what literature the AC is referring to with the statement “It is shown in the literature that models that achieve higher accuracy on ImageNet are not necessarily better at other tasks since they are over-optimized for ImageNet.” Previous literature that measures relationships between ImageNet accuracy and accuracy on downstream tasks is nuanced, and we designed our experiments with previous findings in mind. Kornblith et al. (2019) showed that improvements on ImageNet caused by differences in architecture transfer to other tasks but improvements caused by regularization don’t. We observe a different pattern for our experiments. Both Recht et al. (2019) and Toari et al. (2020) find a strong positive correlation between in-distribution ImageNet accuracy and out-of-distribution accuracy. This is not what we find. We extensively discuss relationships between our findings and previous work in this area in our related work section.
> > >
> > > The conclusion that “contrasting positive against negative examples rather than solely using positive examples improves alignment with human similarity judgments” exclusively relates to our analysis of self-supervised learning models and not to our analysis of ImageNet models that were trained in a supervised manner. We find that self-supervised learning models trained with a contrastive learning objective such as MoCo and SimCLR show a higher degree of alignment than other self-supervised learning models (without such an objective) both zero-shot and after linear probing. We observe this result for both the THINGS triplet odd-one-out data and the human similarity judgments obtained from the multi-arrangement task (see our general response). We are making a factual statement based on the results we observe, and thus we don’t understand the above concern.
> > >
> > > Again, the goal of our paper has not been to examine whether humans or neural nets perform better on a task with ground-truth labels for individual image examples. We wanted to understand which factors drive the degree of alignment with human similarity judgments for which different tasks used in the cognitive sciences (e.g., triplet odd-one-out task) are “just” a means to an end.

---

> > > > ### Author Response · Authors · 2022-12-06
> > > > **Author response (4/4)**
> > > >
> > > > **References**
> > > >
> > > > Cichy, R. M., Kriegeskorte, N., Jozwik, K. M., van den Bosch, J. J., & Charest, I. (2019). The spatiotemporal neural dynamics underlying perceived similarity for real-world objects. Neuroimage, 194, 12-24.\
> > > > Donahue, J., Jia, Y., Vinyals, O., Hoffman, J., Zhang, N., Tzeng, E., & Darrell, T. (2014, January). Decaf: A deep convolutional activation feature for generic visual recognition. In International conference on machine learning (pp. 647-655). PMLR.\
> > > > Hebart, M. N., Dickter, A. H., Kidder, A., Kwok, W. Y., Corriveau, A., Van Wicklin, C., & Baker, C. I. (2019). THINGS: A database of 1,854 object concepts and more than 26,000 naturalistic object images. PloS One, 14(10), e0223792.\
> > > > Hebart, M. N., Zheng, C. Y., Pereira, F., & Baker, C. I. (2020). Revealing the multidimensional mental representations of natural objects underlying human similarity judgements. Nature Human Behaviour, 4(11), 1173-1185.\
> > > > Peterson, Joshua C., Joshua T. Abbott, and Thomas L. Griffiths. "Evaluating (and improving) the correspondence between deep neural networks and human representations." Cognitive science 42.8 (2018): 2648-2669.\
> > > > Kornblith, S., Shlens, J., & Le, Q. V. (2019). Do better ImageNet models transfer better?. In Proceedings of the IEEE/CVF conference on computer vision and pattern recognition (pp. 2661-2671).\
> > > > Razavian, A. S., Azizpour, H., Sullivan, J., & Carlsson, S. (2014). CNN features off-the-shelf: an astounding baseline for recognition. In Proceedings of the IEEE conference on computer vision and pattern recognition workshops (pp. 806-813).\
> > > > Recht, B., Roelofs, R., Schmidt, L., & Shankar, V. (2019, May). Do ImageNet classifiers generalize to ImageNet?. In International Conference on Machine Learning (pp. 5389-5400). PMLR.\
> > > > Taori, R., Dave, A., Shankar, V., Carlini, N., Recht, B., & Schmidt, L. (2020). Measuring robustness to natural distribution shifts in image classification. Advances in Neural Information Processing Systems, 33, 18583-18599.\
> > > > Yamins, D. L., Hong, H., Cadieu, C. F., Solomon, E. A., Seibert, D., & DiCarlo, J. J. (2014). Performance-optimized hierarchical models predict neural responses in higher visual cortex. Proceedings of the national academy of sciences, 111(23), 8619-8624.

---

### Decision · Program_Chairs · 2023-01-20

**Decision:**

Accept: poster

**Justification For Why Not Higher Score:**

As explained above there are some issues in the experiments and conclusions of the paper.

**Justification For Why Not Lower Score:**

The paper presents a valuable study.

**Metareview: Summary, Strengths And Weaknesses:**

The paper studies the alignment of human vision and neural network models via the odd-one-out task (among a triplet of images, choosing the one that is different from the other two). The paper provides an extensive study and reports the results in various settings including different model capacities, using models that are trained on language data, etc.

Strengths:
- The type of analysis that the paper performs is quite interesting.
- The paper is very well written.

Weaknesses:
- The experiments and conclusions have some issues (explained below).


**Note From Pc:**

if the above contains the word "oral" or "spotlight" please see: "oral" presentation means -> notable-top-5% and "spotlight" means -> notable-top-25%. As stated in our emails, we are disassociating presentation type from AC recommendations

**Summary Of Ac-Reviewer Meeting:**

The paper initially received one reject and two borderline ratings. So, the AC organized a virtual meeting with the reviewers to discuss the paper. Only reviewer u84n (the most positive reviewer) participated in the meeting despite multiple reminders. The AC read the paper, the reviews, and the rebuttal carefully. There are a few important issues:

- The dataset used for the analysis has 1854 categories, while most of the models in the paper are trained on ImageNet, which has only 334 classes in common with that dataset (less than 20%). So, the ImageNet-based models have not seen about 80% of the categories during training. Comparing humans with a model that has not seen 80% of those objects is not fair. Basically, the paper compares in-distribution evaluation with out-of-distribution evaluation.


- The maximum achievable odd-one-out-accuracy is about 67%. That means even humans cannot do the task correctly about 33% of the time. Therefore, most of the rankings and conclusions are not reliable due to this large error margin.


- Comparing CLIP models with ImageNet-based models is unfair. CLIP models have seen many more categories, and there might be a higher overlap between the categories in the Things dataset. The better performance of CLIP should not be attributed to language/text. They are trained on data that includes more categories of the Things dataset.


- It is shown in the literature that models that achieve higher accuracy on ImageNet are not necessarily better at other tasks (for example, Kotar et al., ICCV 2021) since they are over-optimized for ImageNet. So, conclusions such as “contrasting positive against negative examples rather than solely using positive examples improves alignment with human similarity judgments.” are not necessarily correct. The models in the paper are used for evaluation of out-of-distribution generalization (generalization to 1520 unseen categories). So, it is expected that a model highly optimized for ImageNet does not generalize well.

Reviewer u84n also agreed with these points. The AC gave a chance to the authors to respond to these issues since they were not raised in the initial reviews. The AC read the provided responses carefully. Some of the responses are not convincing. For example, the response says "The goal of this paper was not to examine whether humans or neural networks perform better on the triplet odd-one-out task". While this is true, the main metric in this paper is odd-one-out accuracy. If humans (i.e. oracle) fail at finding the less similar image in one third of the triplets, it means the data is ambiguous and the accuracies are not reliable for such analysis.

The AC is supportive of these types of studies and also appreciates the effort behind this work. So, despite the issues, the AC is going to recommend acceptance. However, the AC strongly encourages the authors to address these issues in the main text.

---

> ### Author Response · Authors · 2023-02-05
> **Author response**
>
> We have updated our paper with results on two additional human similarity judgment datasets collected by two different groups. The results on these datasets are consistent with the results on the THINGS dataset that we analyzed in our original submission. On these datasets, as on THINGS, we observe that model scale and architecture have no impact on alignment whereas training dataset and objective function have strong effects. We reiterate that one of these datasets uses images from ImageNet and so should overlap with the class distribution of ImageNet.
>
> The AC has raised a set of concerns in their metareview to which we have already responded. The AC then states that “some of the responses are not convincing,” but the only point the AC discusses relates to the reliability of our results. Our updated paper shows that our findings are replicable across datasets, which should eliminate any remaining doubt regarding their reliability. If the AC has further concerns, we are happy to engage in further discussion and/or update our paper accordingly.